# C/EBPδ drives interactions between human MAIT cells and endothelial cells that are important for extravasation

Chang Hoon Lee[1†‡], Hongwei H Zhang[1†], Satya P Singh[1], Lily Koo[2§], Juraj Kabat[2], Hsinyi Tsang[1#], Tej Pratap Singh[1], Joshua M Farber[1*]

[1]Inflammation Biology Section, Laboratory of Molecular Immunology, National Institute of Allergy and Infectious Diseases, National Institutes of Health, Bethesda, United States; [2]Biological Imaging Section, Research Technologies Branch, National Institute of Allergy and Infectious Diseases, National Institutes of Health, Bethesda, United States

*For correspondence:
jfarber@niaid.nih.gov

[†]These authors contributed equally to this work

Present address: [‡]Drug Discovery Division, Korea Research Institute of Chemical Technology, Daejeon, Korea; [§]Center for Devices and Radiological Health, U.S. Food and Drug Administration, Maryland, United States; [#]Cancer Informatics Branch, National Cancer Institute, Rockville, United States

Competing interests: The authors declare that no competing interests exist.

**Abstract** Many mediators and regulators of extravasation by bona fide human memory-phenotype T cells remain undefined. Mucosal-associated invariant T (MAIT) cells are innate-like, antibacterial cells that we found excelled at crossing inflamed endothelium. They displayed abundant selectin ligands, with high expression of *FUT7* and *ST3GAL4*, and expressed CCR6, CCR5, and CCR2, which played non-redundant roles in trafficking on activated endothelial cells. MAIT cells selectively expressed CCAAT/enhancer-binding protein delta (C/EBPδ). Knockdown of C/EBPδ diminished expression of *FUT7*, *ST3GAL4* and *CCR6*, decreasing MAIT cell rolling and arrest, and consequently the cells' ability to cross an endothelial monolayer in vitro and extravasate in mice. Nonetheless, knockdown of C/EBPδ did not affect *CCR2*, which was important for the step of transendothelial migration. Thus, MAIT cells demonstrate a program for extravasation that includes, in part, C/EBPδ and C/EBPδ-regulated genes, and that could be used to enhance, or targeted to inhibit T cell recruitment into inflamed tissue.
DOI: https://doi.org/10.7554/eLife.32532.001

## Introduction

The phenotypes and functions of peripheral T cells are intimately linked to the cells' localization and patterns of migration (*Masopust and Schenkel, 2013*). For memory-phenotype T cells, these relationships were the basis of the dichotomous schema describing central ($T_{CM}$) and effector ($T_{EM}$) memory cells (*Sallusto et al., 1999*). The relationships among position, migration and function have also been emphasized in recent studies characterizing tissue resident memory T cells and recirculating memory T cells (*Bromley et al., 2013*; *Fan and Rudensky, 2016*; *Gerlach et al., 2016*; *Masopust and Picker, 2012*; *Masopust and Schenkel, 2013*). Elegant experiments in mice have shown that resident memory T cells can have critical roles in protection at barrier sites (*Jiang et al., 2012*). Less well defined have been the cells within the memory-phenotype population that can be recruited from the blood to an inflamed site in the very early stages of an immune response (*Masopust and Picker, 2012*). Nonetheless, these 'first responders' can play essential roles in tissue defense (*Kohlmeier et al., 2008*; *Maloy et al., 2000*; *Wakim et al., 2008*).

The recruitment of leukocytes to tissue from blood has been described within the longstanding paradigm, derived primarily from studies of neutrophils, of a multistep process of the cells' rolling, followed by arrest, crawling, and diapedesis (*Ley et al., 2007*). An abundance of data support roles for selectins and their ligands in rolling, and for stimulated chemoattractant receptors and consequent integrin activation in firm arrest and extravasation (*Springer, 1994*). For trafficking of $T_{EM}$ cells

**eLife digest** Lymphocytes are a type of cell found in the blood that can detect and fight infections: in particular, some of them can leave the bloodstream to enter infected or inflamed tissues. To do so, these lymphocytes use proteins on their surface to roll along the inside wall of the blood vessel; then they stick to this wall and finally they pass through it. For some types of lymphocytes the details of this mechanism – such as precisely which surface proteins are necessary – remain unclear.

Here, Lee et al. collect human lymphocytes from the blood of healthy donors, and they identify a subgroup of lymphocytes, called MAIT cells, that are particularly good at moving from blood to infected or inflamed tissues, and further experiments reveal the types of surface proteins that help them do so.

Some of these proteins, for example selectin ligands, are important so the MAIT cell can roll on the wall of the blood vessel. Others, like CCR6, are essential for the cell to stop rolling and stick to the wall. Lee et al. also identify C/EBPδ, a regulatory protein inside the MAIT cell that controls how these other two types of proteins are produced. Finally, Lee et al. show that additional proteins, such as CCR2, are necessary for the lymphocyte to cross the vessel wall.

The proteins that help lymphocytes move from blood to tissues represent important targets to fight diseases. For example, blocking these proteins could prevent lymphocytes from invading and damaging healthy tissues, which happens in autoimmune diseases like multiple sclerosis. Alternatively, manipulating these proteins could help to engineer lymphocytes that can invade and kill tumor tissues in cancers.

DOI: https://doi.org/10.7554/eLife.32532.002

to sites of inflammation, P- and E-selectins on endothelial cells and their glycosylated counterligands on T cells mediate rolling, which initiates the process (*Ley and Kansas, 2004*; *Sperandio et al., 2009*). The best characterized ligands for P- and E-selectin are proteins, such as PSGL-1 (mouse and human), CD43 (mouse and human), ESL-1(mouse), and CD44 (mouse) that bear sugars containing sialyl Lewis$^x$ (sLe$^x$), a tetrasaccharide consisting of N-acetylglucosamine, fucose, galactose, and sialic acid (*Mondal et al., 2013*; *Phillips et al., 1990*). The ability of leukocytes to display selectin ligands is determined primarily by the expression of glycosyltransferases responsible for synthesizing the appropriate glycoforms. These enzymes include core 2 β1,6-N-acetylglucosaminyltransferases, α1,3-fucosyltransferases, and α2,3-sialyltransferases (*Ley and Kansas, 2004*; *Sperandio et al., 2009*). Although the necessary enzymes and a full complement of selectin ligands are expressed constitutively on myeloid cells, expression on lymphocyte populations is heterogeneous and can be affected by cellular activation and differentiation (*Austrup et al., 1997*; *Ley and Kansas, 2004*; *Wagers et al., 1998*). Studies of mouse CD4$^+$ T cells and/or cell lines have shown high preferential expression of selectin ligands on Th1 versus Th2 cells (*Austrup et al., 1997*; *Blander et al., 1999*), and counter-regulation of the α1,3-fucosyltransferase gene *Fut7* by T-bet and GATA-3 (*Chen et al., 2006*); and *Fut7* can be induced in mouse CD4$^+$ T cells in response to a number of cytokines, including IL-12 and TGF-β1(*Ebel et al., 2015*; *Ebel and Kansas, 2016*). Little else is known about the molecular mechanisms regulating the expression of these glycosyltransferases in T cells.

Selectin-mediated rolling allows leukocytes to sample the endothelium for seven-transmembrane domain receptor agonists, principally chemokines, and for ligands, such as VCAM-1, MAdCAM-1, and ICAM-1, for the α4β1, α4β7, and β2 integrins, respectively (*Springer, 1994*). Although signals induced by selectin ligands on neutrophils can yield an integrin conformation sufficient to support integrin-mediated rolling (but not firm arrest), this is not observed for lymphocytes (*Alon and Ley, 2008*). Moreover, except for integrins on recently activated/effector T cells (*Shulman et al., 2011*), integrin activation that is sufficient to induce firm arrest under flow requires chemoattractant receptor-transduced signals (*Alon and Ley, 2008*). Chemokine receptors not only induce integrin activation and leukocyte arrest, but also directly mediate transendothelial migration (TEM) (*Cinamon et al., 2001*; *Shulman et al., 2011*). Among the 19 G-protein-coupled chemokine receptors, only two, CXCR4 and CCR7, are expressed on all naive T cells, whereas T cells with the

effector/memory phenotype can express these and most of the remaining chemokine receptors, resulting in a high degree of combinatorial diversity (*Bachelerie et al., 2014*). The expanded repertoire of chemokine receptors on these cells confers the potential to traffic to and within the wide range of inflammatory sites generated during host defense and injury. There is, however, relatively little understanding of how multiple chemokine receptors can cooperate to provide the functions required by specific T cell subsets, and how the expression of selectin ligands, chemokine receptors and integrins are co-regulated on memory-phenotype T cells in order to confer the ability to extravasate efficiently.

Within the migratory T cell population, the initial cells to enter inflamed tissue should share a $T_{EM}$ phenotype, including not only MHC class I/II restricted cells, but also innate-like T cell such as blood-borne subsets of $\gamma/\delta$ T cells (*Hayday, 2000*), and mucosal-associated invariant T (MAIT) cells (*Gapin, 2014*). In our previous studies, we characterized the subset of human CD4$^+$ T cells co-expressing the chemokine receptors CCR5 and CCR2 (*Zhang et al., 2010*). These cells also express multiple inflammation-associated chemokine receptors and have features of a stable population of highly differentiated cells well equipped to serve as early responding $T_{EM}$. In extending these observations to CD8$^+$ T cells, as described below, we found that most human CD8$\alpha^+$CCR2$^+$ T cells were MAIT cells. MAIT cells are innate-like T cells that express V$\alpha$7.2-J$\alpha$33 (TRAV1-2-TRAJ33 according to the IMGT/GENE-DB nomenclature (*Giudicelli et al., 2005*) as part of a semi-invariant TCR (*Franciszkiewicz et al., 2016*; *Porcelli et al., 1993*), and recognize bacterial metabolites of riboflavin in the context of the non-polymorphic MR1 (*Kjer-Nielsen et al., 2012*). Under homeostatic conditions, MAIT cells are found in the intestinal lamina propria and liver, and represent a significant percentage of CD8$^+$, memory-phenotype T cells in human blood. MAIT cells exhibit potent antibacterial activity and accumulate at sites of bacterial infections (*Gold et al., 2010*; *Le Bourhis et al., 2010*; *Meierovics et al., 2013*). MAIT cells may also have roles in immune-mediated diseases (*Hinks, 2016*). Although based on their expression of chemokine receptors and other surface markers, MAIT cells have been characterized as tissue-homing (*Dusseaux et al., 2011*), there are no detailed studies of their trafficking behavior.

Human MAIT cells have generally been identified by their co-expression of TCRV$\alpha$7.2 and the NK cell marker CD161 (*Franciszkiewicz et al., 2016*). The discovery of the MR1-bound ligands for the MAIT cell TCRs has allowed for the development of MR1 tetramers (*Reantragoon et al., 2013*). Although these tetramers represent a significant new tool for studying MAIT cells, recent data show that the tetramers of MR1 bound to riboflavin- and folate-derived ligands can also identify a heterogeneous collection of V$\alpha$7.2$^-$ non-MAIT cells, thereby defining a broader population including both MAIT cells and 'atypical' V$\alpha$7.2$^-$ MR1-restricted T cells (*Gherardin et al., 2016*; *Meermeier et al., 2016*). The vast majority of MAIT cells in human blood are CD8$\alpha^+$ (many of which are also CD8$\beta^{low}$), and other than expression of CD8, no differences have been noted between CD8$^+$ and CD8$^-$ MAIT cells (*Franciszkiewicz et al., 2016*; *Walker et al., 2012*). For the sake of brevity, because the work described below deals only with the MAIT cells that are CD8$\alpha^+$, we will use 'MAIT' in place of 'CD8$\alpha^+$ MAIT'.

In our current work, we found that MAIT cells were highly efficient at extravasation across inflamed endothelium. We also discovered that MAIT cells selectively and highly express the bZIP transcription factor C/EBP$\delta$. siRNA-mediated knockdown of *C/EBP$\delta$* showed that C/EBP$\delta$ contributed to the expression of glycosyltransferases/selectin ligands as well as CCR6 on MAIT cells, and consequently was required for optimal rolling and arrest of these cells on activated endothelial cells. Although these effects led to a decrease in the number of MAIT cells crossing the endothelium, knocking down *C/EBP$\delta$* had no separate effect on the final and critical step of transendothelial migration - which depended on CCR2. Taken together, our data show that MAIT cells are efficient at trafficking across inflamed endothelium due to a coordinated program, regulated in part by C/EBP$\delta$, that controls genes encoding proteins of disparate activities, each contributing to the migratory phenotype.

# Results

## CD161 and CCR6 are co-expressed on CD8α⁺TCRVα7.2⁺ cells

In expanding our previous studies of CCR2- and CCR5-expressing subsets of CD4⁺ T cells (*Zhang et al., 2010*), we characterized CD8α⁺CCR2⁺ T cells from human peripheral blood. We found that approximately 80% of the CD8α⁺CCR2⁺ T cells expressed TCRVα7.2 and CD161 (*Figure 1A*), which identify them as MAIT cells (*Martin et al., 2009*), and approximately 75% of MAIT cells were CCR2⁺. For additional analysis of chemokine receptor expression on MAIT cells, we divided the CD8α⁺ T cell population into naive and memory-phenotype cells, and further divided the memory-phenotype cells into CCR6⁻ conventional (non-MAIT) cells, CCR6⁺ conventional (non-MAIT) cells, CCR2⁻/low MAIT cells, and CCR2⁺ MAIT cells (see *Figure 2—figure supplement 1A*, below). In addition to CCR2, we found that MAIT cells also prominently expressed CXCR4, CXCR6, CCR5 and CCR6, but lacked CCR7 (*Figure 1B*), consistent with published data (*Dusseaux et al., 2011*). CCR6 and CD161 are co-expressed on Th17 cells (*Cosmi et al., 2008*), and we found that CCR6 and CD161 marked virtually identical cells within the CD8α⁺TCRVα7.2⁺ subset, so that CCR6 and CD161 could be used interchangeably for identifying MAIT cells (*Figure 1C*).

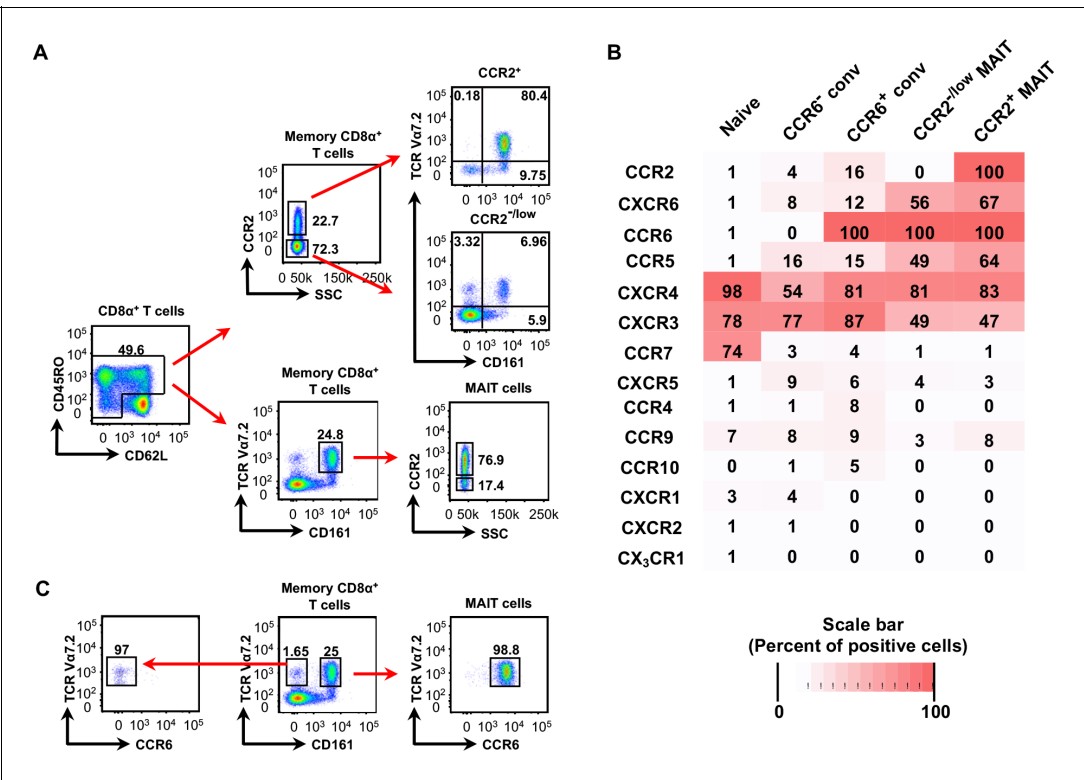

**Figure 1.** Most CD8α⁺CCR2⁺ cells are MAIT cells and all CD8α⁺ MAIT cells express CCR6. (**A**) Expression of CCR2 on CD8α⁺ memory-phenotype cells and frequencies of TCR V$_\alpha$7.2⁺ CD161⁺ (MAIT) cells within the CCR2⁺ and CCR2⁻ subsets (top), and expression of CCR2 within the MAIT cell population (bottom). Cells are shown from one representative of more than 30 donors. (**B**) Expression of chemokine receptors on naive CD8α⁺ T cells and memory-phenotype CD8α⁺ T cells divided into the following subsets: TCR V$_\alpha$7.2⁻ CCR6⁻ (CCR6⁻ conventional), TCR V$_\alpha$7.2⁻ CCR6⁺ (CCR6⁺ conventional), TCR V$_\alpha$7.2⁺ CCR6⁺ CCR2⁻/low (CCR2⁻/low MAIT), and TCR V$_\alpha$7.2⁺ CCR6⁺ CCR2⁺ (CCR2⁺ MAIT); shades of red reflect percentages of cells positive for each chemokine receptor as determined by flow cytometry and as displayed in each box. Data are averaged from cells from nine donors. (**C**) Expression of CCR6 on MAIT cells (right) and TCR V$_\alpha$7.2⁺ CD161⁻ cells (left). Numbers are percentages of cells within a quadrant or demarcated region. Cells are shown from one representative of six donors.

DOI: https://doi.org/10.7554/eLife.32532.003

The following source data is available for figure 1:

**Source data 1.** Data for *Figure 1B*, flow cytometry results for cells from individual donors.
DOI: https://doi.org/10.7554/eLife.32532.004

## Increased rolling, arrest, and TEM of CCR2⁺ MAIT cells on activated endothelial cells

Given the pattern of chemokine receptor expression on MAIT cells, our earlier data on CD4⁺CCR5⁺-CCR2⁺ T cells as potential 'first responders' (*Zhang et al., 2010*), and data from others identifying CCR5 and CCR2 as important receptors for TEM on effector/activated T cells (*Shulman et al., 2011*), we considered whether MAIT cells - and in particular their CCR2⁺ subset - might exhibit efficient extravasation in the context of inflammation. We began the analysis using flow chamber assays with human umbilical vein endothelial cell (HUVEC) monolayers pre-treated overnight with TNFα. For studying T-cell-endothelial cell interactions, we purified subsets of CD8α⁺ T cells from peripheral blood using cell sorting, divided into MAIT cell and conventional cell subsets as described above and shown in *Figure 2—figure supplement 1A*.

The staining with the anti-CCR2 antibody characteristically showed a continuum between CCR2⁻ and CCR2⁺ MAIT cells (*Figure 1A* and *Figure 2—figure supplement 1A*), and the limited number of CCR2⁻ MAIT cells often required using a relaxed gate for obtaining sufficient numbers of these cells for trafficking experiments. Consequently, the 'CCR2⁻' MAIT cells typically contained CCR2^low cells, and we have designated this subset as CCR2^{-/low} MAIT cells accordingly. Even when using a severe gate for obtaining CCR2⁻ and CCR2⁺ MAIT cells for other studies, such as analyzing gene expression, expression of *CCR2* mRNA was found in the CCR2⁻ MAIT cells (see Figure 7I below), supporting the use of the CCR2^{-/low} designation. We separated the CCR6⁺ from the CCR6⁻ conventional memory-phenotype cells for purposes of comparison with MAIT cells, since all MAIT cells are CCR6⁺ (*Figure 1C*). It is notable that the CCR2⁺ MAIT cells expressed the highest levels of CCR6 among the memory-phenotype subsets divided in this way (*Figure 2—figure supplement 1B*).

T cell subsets were introduced into the flow chambers at 0.75 dyn/cm² for 4 min after which shear stress was increased to 5 dyn/cm² and data on cell numbers were collected for 16 min. Naive CD8α⁺ T cells did not adhere to the HUVECs, and among the memory-phenotype cells there was, generally, a pattern of progressive increase in numbers of cells rolling, arrested, and transmigrating going from CCR6⁻ conventional to CCR6⁺ conventional to CCR2^{-/low} MAIT to CCR2⁺ MAIT cells (*Figure 2A*). Although in our assays we detected TEM using differential interference contrast (DIC) microscopy, we were able to confirm that migration under the endothelial cell monolayer was occurring by using confocal microscopy with CFSE-stained CCR2⁺ MAIT cells and cell tracker Red CMTPX-stained HUVECs (*Figure 2—figure supplement 2A*). Because rolling, arrest, and TEM are sequential, where arrest is predicated on rolling and TEM predicated on arrest, we calculated numbers of cells showing arrest and TEM as percentages of cells rolling or arresting, respectively. The data showed that the MAIT cells were most efficient at the step of arresting, and that the CCR2⁺ MAIT cells were particularly effective in the final step of TEM (*Figure 2B*).

δIn addition, among those cells that transmigrated, the cells within the CCR2⁺ MAIT subset took the shortest time between arrest and completion of transmigration. As an example, as was seen for cells from one donor, two out of three CCR2⁺ MAIT cells transmigrated in 23 s, and 1 min 55 s after arrest, whereas the CCR6⁺ conventional and CCR2^{-/low} MAIT cells took 3 min 15 s and 2 min 42 s, respectively (*Figure 2—figure supplement 2B*). The rapid transmigration of the CCR2⁺ MAIT cells can be seen in *Video 1*, in which the white arrowheads mark cells at the initiation of TEM, and pooled data from five donors demonstrate significant differences among the times between arrest and TEM for the transmigrating cells from the CCR6⁺ conventional cells, CCR2^{-/low} MAIT cells and CCR2⁺ MAIT cells (*Figure 2—figure supplement 2C*). Overall, these patterns suggested coordinated and co-regulated capabilities in rolling, firm arrest, and TEM among the memory-phenotype subsets, with the CCR2⁺ MAIT cells best equipped for extravasation.

## CCR2⁺ MAIT cells traffic to inflamed tissue

In order to assess the ability of these subsets to traffic in vivo, we injected purified, CFSE-labeled cells into the left hearts of mice and, using flow cytometry, analyzed labeled cells remaining in inflamed (and non-inflamed) ears within a few minutes after injection. In these experiments, and some additional experiments that follow, the limitations in cell numbers discussed above prevented us from evaluating the CCR2^{-/low} MAIT subset. We detected retention of CCR2⁺ MAIT cells, but not cells from the other subsets, in ears that had had prior intradermal injection of TNFα and IL-1β (*Figure 2C*). We failed to detect any of these cells in non-inflamed ears. Confocal microscopy of ears

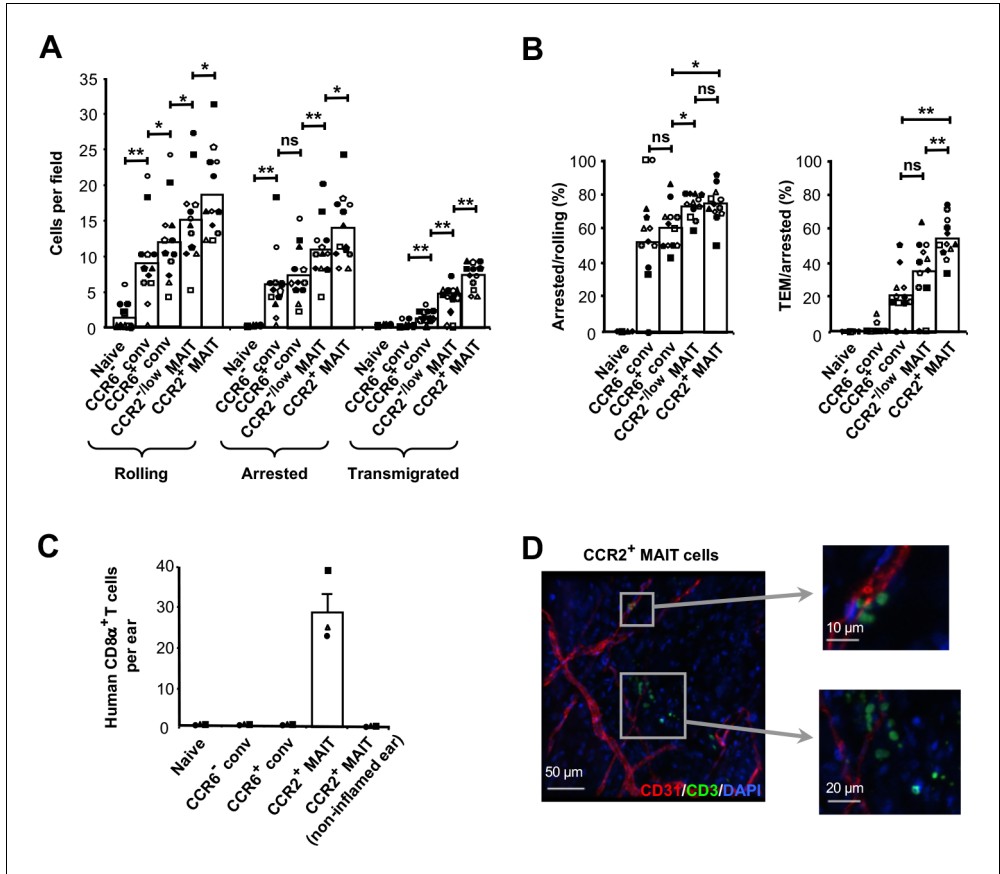

**Figure 2.** CCR2+ MAIT cells are highly efficient at TEM. (**A**) Numbers of cells rolling, arrested, and transmigrated on TNFα-activated HUVECs for CD8α+ T cells divided into subsets as in *Figure 1B* and shown in *Figure 2—figure supplement 1*. (**B**) Percentages of rolling and arrested cells that arrested and transmigrated, respectively, calculated from the data in (**A**). Bars in (**A**) and (**B**) show means, and data are from cells from 12 donors, each identified by a unique symbol. The p values were calculated using the Wilcoxon signed rank test. (**C**) Numbers of CFSE-labeled human CD8α+ T cells recovered from TNFα/IL-1β-injected mouse ears after intra-cardiac injections of T cell subsets. Bars show means and SEMs from three experiments, each with one mouse injected with $1 \times 10^6$ cells from each subset. (**D**) Confocal microscopy of ear sheets prepared from mice after injection of purified CCR2+ MAIT cells as in (**C**) and stained for human T cells (anti-human CD3, green), mouse endothelial cells (anti-murine CD31, red) and nuclei (DAPI, blue). Squares indicate areas magnified at right. Images are representative of four experiments. (**A** and **B**) ns, not significant; *, $p<0.05$; **, $p<0.01$.

DOI: https://doi.org/10.7554/eLife.32532.005

The following source data and figure supplements are available for figure 2:

**Source data 1.** Data for *Figure 2A, B*, flow chamber results for cells from individual experiments.
DOI: https://doi.org/10.7554/eLife.32532.008
**Figure supplement 1.** CCR2+ MAIT cells are CCR6high.
DOI: https://doi.org/10.7554/eLife.32532.006
**Figure supplement 2—Source data 1.** Data for *Figure 2—figure supplement 2C*, flow chamber results for individual cells.
DOI: https://doi.org/10.7554/eLife.32532.009
**Figure supplement 2.** CCR2+ MAIT cells undergo rapid TEM.
DOI: https://doi.org/10.7554/eLife.32532.007

after immuno-staining showed that labeled cells had extravasated into tissue (*Figure 2D* and *Video 2*).

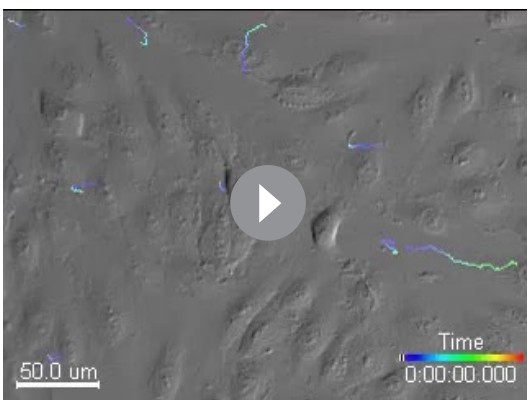

**Video 1.** CCR2[+] MAIT cells transmigrate soon after arresting on TNFα-activated HUVECs.
DOI: https://doi.org/10.7554/eLife.32532.010

## Rolling correlates with levels of selectin ligands and glycosyltransferases

In order to understand the basis for the differences in rolling frequencies among the T-cell subsets, we analyzed adhesion molecules on the TNFα-treated HUVECs and the T cells. On the HUVECs, we detected up-regulation of E-, but not P-selectin, consistent with published data (*Yao et al., 1996*), and up-regulation of both ICAM-1 and VCAM-1 (*Figure 3A*). Staining the T-cell subsets using E-selectin-Fc and P-selectin-Fc chimeric proteins showed a progressive increase in staining from naive to CCR6[−] conventional to CCR6[+] conventional to CCR2[-/low] MAIT to CCR2[+] MAIT cells (*Figure 3B*). A similar pattern was seen in staining the cells for sLe[x] (*Figure 3C*). Consistent with the critical role for sLe[x] in mediating rolling, we showed that treating cells with an exo-sialidase, which eliminated staining for sLe[x] (*Figure 3—figure supplement 1A*), abolished rolling on TNFα-treated HUVECs (*Figure 3D*).

The differences in levels of selectin ligands among the T cell subsets could be due to differences in expression of the proteins that bear the relevant sugars and/or the degree of the appropriate glycosylation. Levels of surface PSGL-1 and CD43 could not explain the pattern of selectin ligand expression (*Figure 3—figure supplement 1B*). We also checked expression of CD44, even though CD44 has only been shown to be a selectin ligand on mouse neutrophils (*Hidalgo et al., 2007*; *Mondal et al., 2013*). CD44 could also not account for the pattern of selectin ligand expression, although CD44 was found at somewhat higher levels on the CCR6[+] subsets as compared with the naive and CCR6[-] conventional cells. In general, expression of selectin ligands reflects regulation of the genes encoding the limiting glycosyltransferases (*Ley and Kansas, 2004*). We found that patterns of expression of *GCNT1*, encoding core 2 β1,6-N-acetylglucosaminyltransferase-I, *FUT4*, *FUT6*, *FUT9*, *ST3GAL1, 2, 3*, and six were unremarkable (*Figure 3E* and *Figure 3—figure supplement 1C*). However, expression of *FUT7* and *ST3GAL4* were highest in the MAIT cells, and particularly in the CCR2[+] MAIT cell subset (*Figure 3E*), consistent with the MAIT cells' high expression of selectin ligands and the important roles for *FUT7* and *ST3GAL4* in the synthesis of selectin ligands in leukocytes (*Ellies et al., 2002*; *Malý et al., 1996*; *Mondal et al., 2015*).

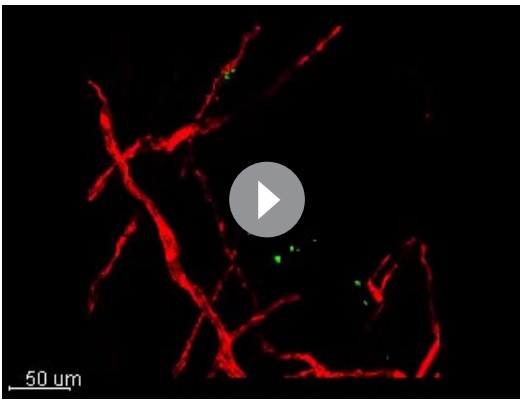

**Video 2.** CCR2[+] MAIT cells extravasate into the inflamed mouse ear.
DOI: https://doi.org/10.7554/eLife.32532.011

## CCR6 mediates firm arrest

Because leukocyte arrest on endothelium typically requires integrin activation in response to signals from chemoattractant receptors, we investigated the contributions of these two classes of proteins to the differences in the behaviors of the T cell subsets. As can be inferred from the expression patterns of integrin subunits (*Figure 4—figure supplement 1*), there were no differences in expression of LFA-1, VLA-4, and $\alpha_4\beta_7$ that could explain the differences in efficiencies of arrest among the subsets. To determine the overall role of chemokine receptors, which couple to $G_{i/o}$ G proteins, we analyzed T cells with and without pretreatment with pertussis toxin. Pertussis toxin had no effect on rolling, reduced the numbers of arrested memory-

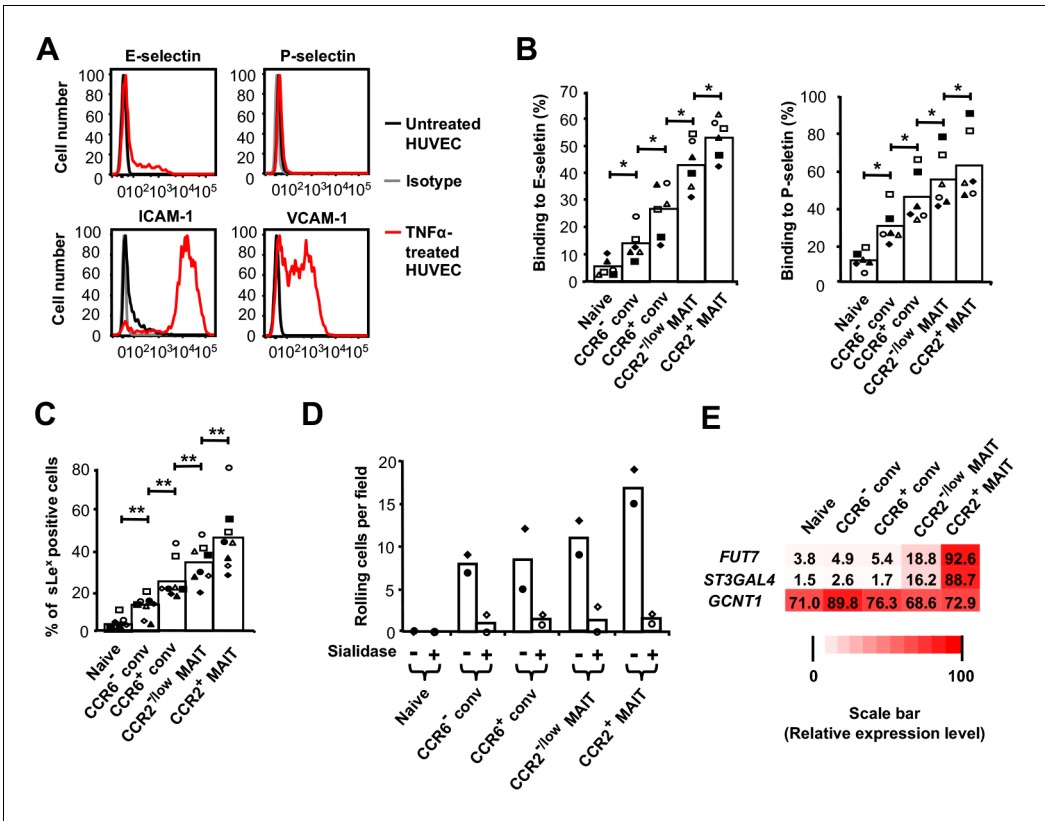

**Figure 3.** Rolling correlates with expression of selectin ligands and *FUT7* and *ST3GAL4*. (**A**) Expression of adhesion molecules on untreated HUVECs and TNFα-treated HUVECs. Staining of untreated HUVECs with the four different antibodies and TNFα-treated HUVECs with isotype-matched antibodies are shown as negative controls. Data are from one representative of three experiments. (**B**) Percentages of cells binding to the E-selectin-Fc and P-selectin-Fc chimeric proteins. CD8α+ T cells were divided into subsets as in *Figure 1B* and shown in *Figure 2—figure supplement 1*. (**C**) Percentages of sLe^X-positive cells within each CD8α+ T cell subset. (**B and C**) Bars show means, and data are from cells from six (**B**) or those six plus two additional (**C**) donors, each identified by a unique symbol within each panel. The p values were calculated using the Wilcoxon signed rank test. (**D**) Numbers of cells rolling per field on TNFα-activated HUVECs for CD8α+ T cell subsets, either untreated or pre-treated with sialidase. Bars show means from cells from two donors as represented by the two symbols. (**E**) Expression of *FUT7*, *ST3GAL4* and *GCNT1* in CD8α+ T cell subsets; shades of red and numbers displayed in each box represent relative levels of expression based on values for $2^{-\Delta CT}$ obtained by real-time RT-PCR. Data are averaged from cells from three donors. (**B and C**) *, p<0.05; **, p<0.01.
DOI: https://doi.org/10.7554/eLife.32532.012

The following source data and figure supplements are available for figure 3:

**Source data 1.** Data for *Figure 3B, C* (flow cytometry results for cells from individual donors), *Figure 3D* (flow chamber results for cells from individual experiments), and *Figure 3E*, (normalized mRNA expression in cells from individual experiments).
DOI: https://doi.org/10.7554/eLife.32532.014

**Figure supplement 1.** Selectin ligands and glycosyltransferases in CD8α+ T cells.
DOI: https://doi.org/10.7554/eLife.32532.013

**Figure supplement 1—source data 1.** Data for *Figure 3—figure supplement 1B*, mRNA expression in cells from individual experiments.
DOI: https://doi.org/10.7554/eLife.32532.015

phenotype cells by approximately 50%, and eliminated TEM (*Figure 4A*).

In order to focus on the chemokine receptors important for arrest and TEM of MAIT cells in the flow chambers, we analyzed the TNFα-treated endothelial cells for expression of the mRNAs for the chemokine ligands of the receptors highly expressed on the MAIT cells. TNFα induced high expression of the genes for the CCR2 ligand CCL2, the CCR5 ligand CCL5, and the CCR6 ligand CCL20,

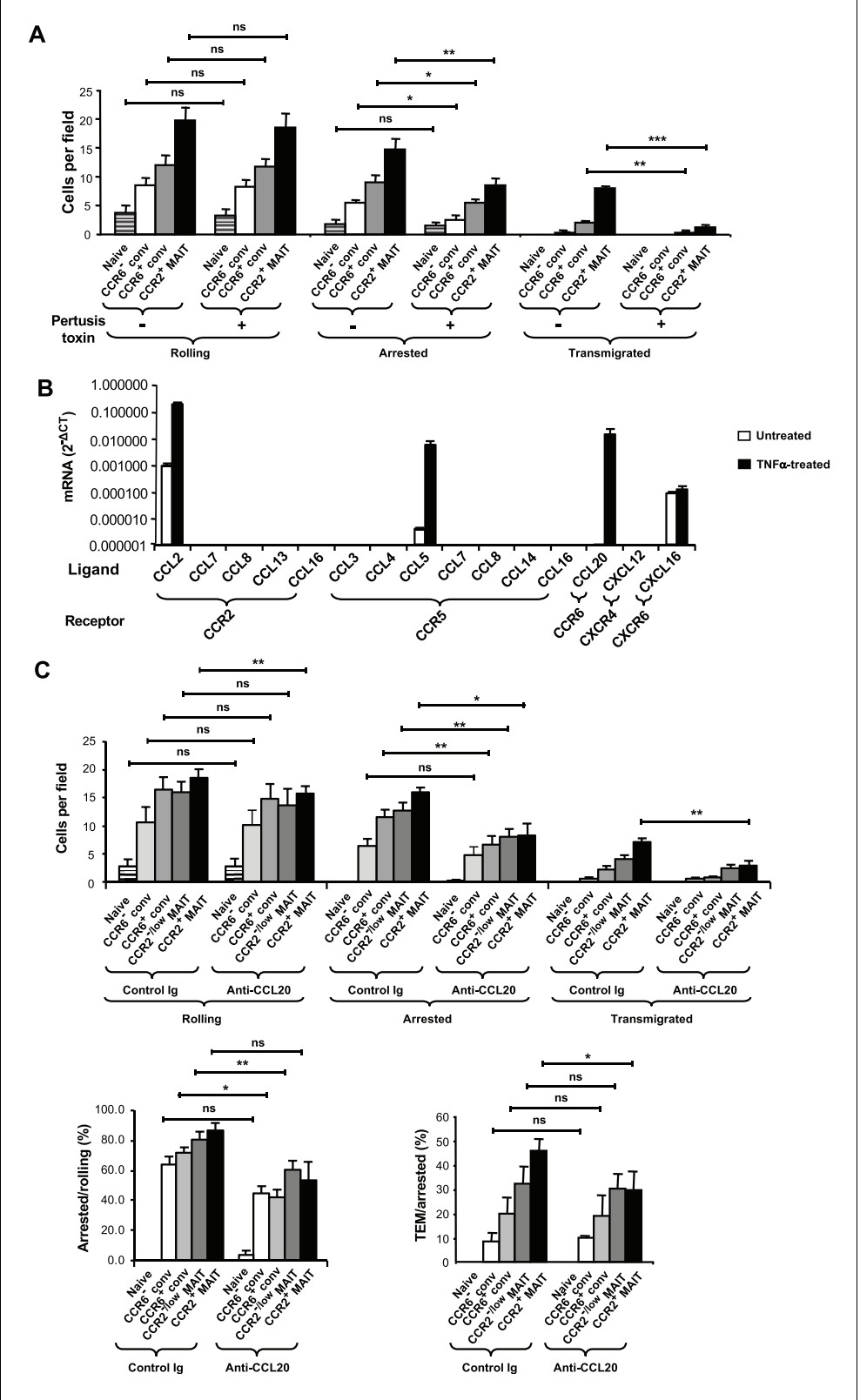

**Figure 4.** CCR6 mediates arrest. (**A**) Numbers of cells rolling, arrested, and transmigrated per field on TNFα-activated HUVECs for CD8α+ T cell subsets, either untreated or treated with pertussis toxin. CD8α+ T cells were divided into subsets as in *Figure 1B* and shown in *Figure 2—figure supplement 1*, except that the CCR2-/low MAIT cells were not studied. (**B**) Expression of mRNAs, normalized to *GAPDH* expression, encoding chemokine

*Figure 4 continued on next page*

*Figure 4 continued*

ligands for the listed receptors in HUVECs either untreated or treated with TNFα. Bars show means and SEMs from two experiments. (C) Numbers of T cells rolling and arrested per field on TNFα-activated HUVEC that had been pre-treated with either control IgG or anti-human CCL20 antibody (top); percentages of rolling and arrested cells that arrested and transmigrated, respectively, calculated from the data shown above. CD8α⁺ T cells were divided into subsets as in *Figure 1B* and shown in *Figure 2—figure supplement 1*. (A and C) Bars show means and SEMs. Data are from cells from four (A) and five (C) donors. The p values were calculated using the paired t test. (A and C) ns, not significant; *, p<0.05; **, p<0.01; ***, p<0.001.
DOI: https://doi.org/10.7554/eLife.32532.016

The following source data and figure supplement are available for figure 4:

**Source data 1.** Data for *Figure 4A, C* (flow chamber results for cells from individual experiments) and *Figure 4B* (mRNA expression in individual experiments).
DOI: https://doi.org/10.7554/eLife.32532.018

**Figure supplement 1.** Expression of integrins does not account for MAIT cells' enhanced arrest on activated endothelial cells.
DOI: https://doi.org/10.7554/eLife.32532.017

whereas CXCL16 was expressed at only a low level that was not augmented by TNFα (*Figure 4B*). We tested a role for CCR6 by using antibody to CCL20. Anti-CCL20 antibody significantly reduced numbers of cells arresting in the three CCR6⁺ subsets. Similarly, reflecting the effects specifically on the step of arresting, anti-CCL20 significantly reduced the percentage of rolling cells that arrested in two of the three CCR6⁺ subsets, but not in the CCR6⁻ cells (*Figure 4C*). Although convincing, the effects of anti-CCL20 were not large. It is notable in this regard that given the results using pertussis toxin, a 50% reduction in arresting cells is the maximum decrease that could have been expected by blocking a Gα$_{i/o}$-coupled receptor such as CCR6.

Our data did show statistically significant differences between control- and anti-CCL20-treated CCR2⁺ MAIT cells in numbers of cells rolling and percentages of arrested cells that underwent TEM. Because similar results were not found for the other two CCR6-expressing subsets (CCR6⁺ conventional and CCR2$^{-/low}$ MAIT cells), we concluded that CCR6 lacked a convincing activity in rolling or TEM of these cells. Taken together, the data suggest a role for CCR6 in arrest of the CCR6-expressing CD8α⁺ memory-phenotype T cells, and little or no effects on the steps of rolling and TEM.

## CCR2 mediates TEM and CCR5 shortens the time between arrest and TEM

We addressed the role of CCR2 in TEM by using the CCR2 antagonist, BMS CCR2 22. BMS CCR2 22 did not diminish the numbers of cells rolling or arrested - the lack of effect on firm arrest ruling out a general inhibition of chemokine receptor signaling. However, blocking CCR2 had a profound effect on TEM of the CCR2⁺ MAIT cells (*Figure 5A*). The CCR2 inhibitor also significantly diminished TEM of the CCR6⁺ conventional cells, consistent with the expression of CCR2 on a subset of these cells (*Figure 2—figure supplement 1A*). The studies using the CCR2 inhibitor were among those experiments in which we did not include CCR2$^{-/low}$ MAIT cells due to the low numbers of these cells that we could obtain from an individual donor. It is notable, however, that significant numbers of cells within the CCR2$^{-/low}$ MAIT cell samples were able to undergo TEM (*Figure 2*). We presume that CCR2 expressed on the CCR2$^{low}$ cells within these samples was responsible for this TEM.

Given the high expression of CCR5 by MAIT cells (*Figure 1B*) and expression of CCL5 by the TNFα-activated endothelial cells, we also investigated a role for CCR5 using the CCR5 antagonist, maraviroc. Blocking CCR5 had no effect on the numbers of cells rolling, arrested, or transmigrated (*Figure 5B*). However, among the transmigrating CCR2⁺ MAIT cells, blocking CCR5 significantly prolonged the time between when the cells initiated crawling and initiated TEM (*Figure 5C*). There was no effect on the time interval between arrest and initiating crawling.

These data on the activity of CCR5 may partly explain the differences in time between arrest and TEM that we measured for the CCR6⁺ conventional cells, CCR2$^{-/low}$ MAIT cells and the CCR2⁺ MAIT cells (*Figure 2—figure supplement 2C*), because we found differences in the levels of CCR5 expression in the order CCR2⁺ MAIT cells > CCR2$^{-/low}$ MAIT cells > the CCR2⁺ subset of CCR6⁺ conventional cells (*Figure 5—figure supplement 1*). For measuring levels of CCR5 on the surface of the

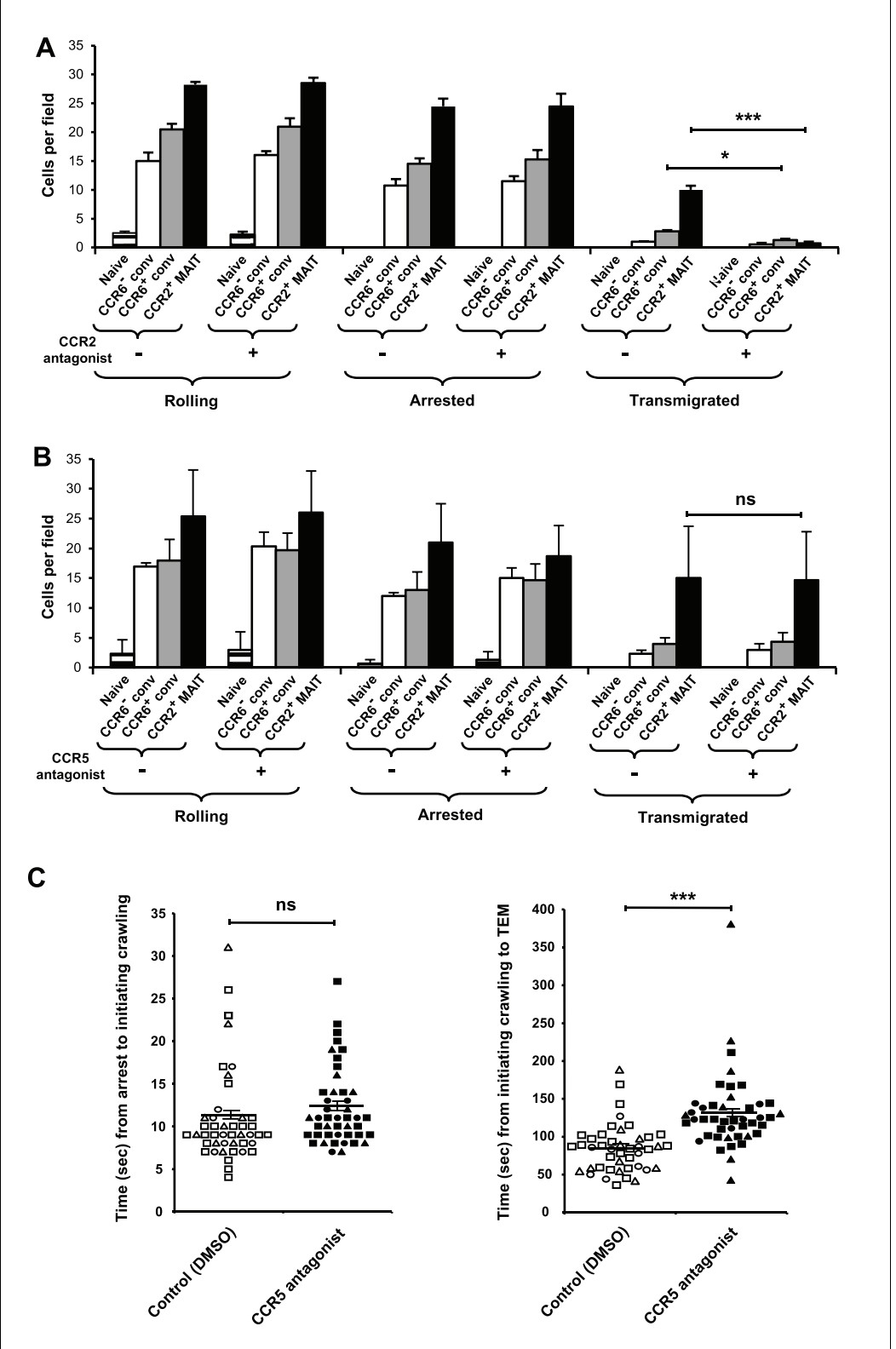

**Figure 5.** CCR2 and CCR5 contribute to TEM. (**A**) Numbers of cells rolling, arrested, and transmigrated per field on TNFα-activated HUVECs for CD8α⁺ T cells, either untreated or treated with a CCR2 antagonist. CD8α⁺ T cells were divided into subsets as in *Figure 1B* and shown in *Figure 2—figure supplement 1*, except that the CCR2$^{-/}$ $^{low}$ MAIT cells were not studied. (**B**) Numbers of cells rolling, arrested, and transmigrated as described in (**A**) for

*Figure 5 continued on next page*

*Figure 5 continued*

cells either untreated or treated with a CCR5 antagonist. (**A and B**) Bars show means and SEMs. Data are from cells from four (**A**) and three (**B**) donors. The p values were calculated using the paired t test. (**C**) Times between arrest and initiating crawling and between initiating crawling and initiating TEM for CCR2$^+$ MAIT cells on TNFα-activated HUVECs, either control-treated (open symbols) or treated with a CCR5 antagonist (closed symbols). Each symbol represents an individual cell; open-ended horizontal lines show means and SEMs. Data are from cells from three donors as represented by the three types of symbols. The p value was calculated using the unpaired t test. ns, not significant; *, p<0.05; ***, p<0.001.

DOI: https://doi.org/10.7554/eLife.32532.019

The following source data and figure supplements are available for figure 5:

**Source data 1.** Data for *Figure 5A, B* (flow chamber results for cells from individual experiments) and *Figure 5C* (flow chamber results for individual cells).

DOI: https://doi.org/10.7554/eLife.32532.021

**Figure supplement 1.** Levels of CCR5 expression differ among CD8α$^+$ subsets.

DOI: https://doi.org/10.7554/eLife.32532.020

**Figure supplement 1—source data 1.** Data for *Figure 5—figure supplement 1B*, flow cytometry results for cells from individual donors.

DOI: https://doi.org/10.7554/eLife.32532.022

CCR6$^+$ conventional cells, we limited the analysis to cells that were CCR2$^+$ because the data in *Figure 5A* indicated that the CCR2$^+$ cells were those undergoing TEM, and therefore were the cells scored in *Figure 2—figure supplement 2C*. For the CCR2$^{-/low}$ MAIT cells, in addition to having levels of CCR5 that were lower than on the CCR2$^+$ MAIT cells, the reduced expression of CCR2 presumably also contributed to their delay in initiating TEM.

## C/EBPδ regulates MAIT cell trafficking

Having determined some of the factors contributing to the trafficking behavior of MAIT cells, we next investigated how expression of these factors might be regulated. In previous, unpublished experiments we had characterized gene expression in the CCR5$^+$CCR2$^+$ subset of CD4$^+$ memory-phenotype T cells that we had studied earlier and discovered that these cells expressed high levels of *CEBPD*, which encodes C/EBPδ. We found that MAIT cells also expressed high levels of *CEBPD* mRNA and C/EBPδ, and that these could be knocked down using *CEBPD* siRNAs (*Figure 6A and B*). We found no selective expression in MAIT cells of the related genes *CEBPA*, *CEBPB*, *CEBPE*, *CEBPG*, and *CEBPZ* (*Figure 6—figure supplement 1*). siRNA-mediated knockdown of *CEBPD* diminished the overall trafficking of MAIT, but not non-MAIT cell subsets in the flow chamber assays (*Figure 6C*). Knockdown of C/EBPδ had a significant effect on the rolling step (*Figure 6C*) and a modest, separate effect on firm arrest of the MAIT cells (percentage of rolling cells undergoing arrest), although the effect on the CCR2$^+$ MAIT cell subset did not reach statistical significance (*Figure 6D*). Importantly, although knocking down C/EBPδ diminished the numbers of MAIT cells crossing the activated endothelial cells through effects on the initial trafficking steps, knocking down C/EBPδ had no consistent, independent effect on TEM (percentage of arrested cells undergoing TEM) in the MAIT cell subsets (*Figure 6D*).

To evaluate a role for C/EBPδ in trafficking in vivo, we co-injected differentially labeled CCR2$^+$ MAIT cells that had been transfected with control or *CEBPD* siRNA into mice whose ears had been injected with TNFα and IL-1β. As compared to controls, significantly fewer cells with knockdown of C/EBPδ could be recovered from the inflamed ears (*Figure 6E*).

## C/EBPδ regulates glycosyltransferases and CCR6

In order to understand the basis of the effects of knocking down C/EBPδ on rolling and arrest, we investigated if knockdown of C/EBPδ affected expression of sLe$^x$/glycosyltransferases and chemokine receptors. Knockdown of C/EBPδ decreased surface levels of sLe$^x$ and decreased expression of *FUT7* and *ST3GAL4* in MAIT, but not in conventional cells (*Figure 7A– C*). Knockdown of C/EBPδ did not decrease expression of *GCNT1* (*Figure 7—figure supplement 1*). Analysis of the 5' flanking regions of *FUT7* and *ST3GAL4* (*Figure 7—figure supplement 2*), identified potential binding sites

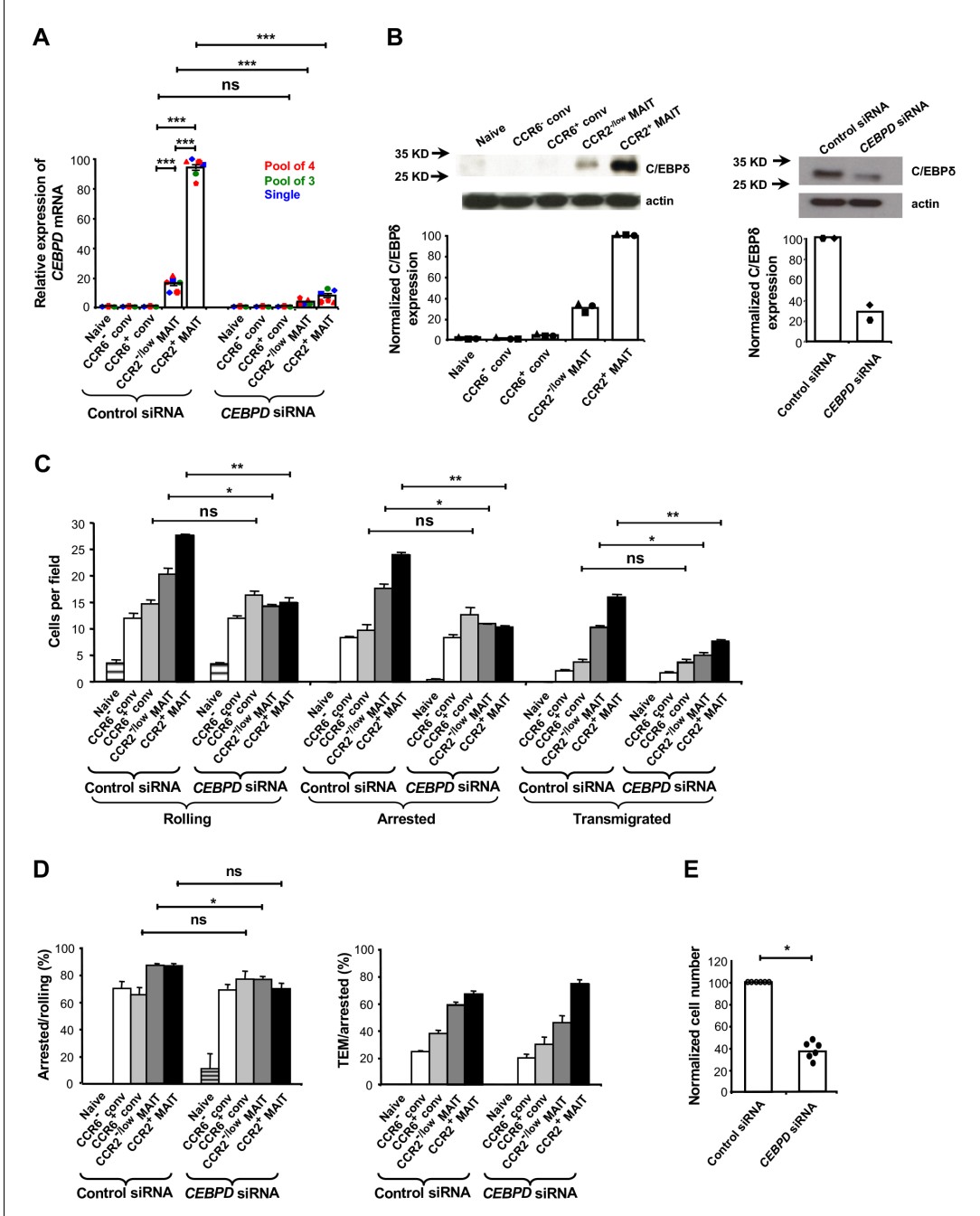

**Figure 6.** C/EBPδ supports rolling and arrest of MAIT cells. (**A**) Relative expression of *CEBPD* mRNA in CD8α⁺ T cells after transfections with control or *CEBPD* siRNA. CD8α⁺ T cells were divided into subsets as in *Figure 1B* and shown in *Figure 2—figure supplement 1*. Each symbol type shows results from an individual donor. Red and green symbols are from experiments using non-overlapping pools of four or three CEBPD siRNAs, respectively, and blue symbols are from experiments using one of the siRNAs from the pool of three. Other experiments in this figure used the pool of four siRNAs and in all experiments using siRNAs, cells were harvested 3–4 days after transfections. Bars show means and SEMs. Data are from cells from six donors. The p values were calculated using the paired t test. (**B**) Expression of C/EBPδ in CD8α⁺ T cell subsets, either untreated (left) or after transfections with control or *CEBPD* siRNA (right). Actin bands demonstrate equal loading and arrows indicate positions of molecular weight markers. Blots are from cells from one representative of three (left) and two (right) donors, with quantification of the blots shown below. (**C**) Numbers of cells rolling, arrested, and transmigrated per field on TNFα-activated HUVECs for CD8α⁺ T cell subsets after transfections with control or *CEBPD* siRNA. (**D**) Percentages of rolling and arrested cells that arrested and transmigrated, respectively, calculated from the data in (**C**). (**C and D**) Bars show means and SEMs. Data are from cells from three donors. The p values were calculated using the paired t test. (**E**) Relative numbers of CCR2⁺ MAIT cells transfected with *CEBPD* versus control siRNAs recovered from TNFα/IL-1β-injected mouse ears eight minutes after intra-cardiac injection of a 1:1

*Figure 6 continued on next page*

*Figure 6 continued*

mixture of differentially labeled cells. Values were normalized to numbers of the cells transfected with control siRNA. Bars show means. Data are from four experiments with a total of six mice. The p value was calculated using the Wilcoxon signed rank test. (**A**, **C**, **D**, and **E**) ns, not significant; *, p<0.05; **, p<0.01; ***, p<0.001.

DOI: https://doi.org/10.7554/eLife.32532.023

The following source data and figure supplements are available for figure 6:

**Source data 1.** Data for *Figure 6A* (normalized mRNA expression in cells from individual experiments), *Figure 6E*, (quantification of Western blots from individual experiments), *Figure 6C, D*, (flow chamber results for cells from individual experiments), and *Figure 6E*, (normalized cell numbers from ears from individual experiments).

DOI: https://doi.org/10.7554/eLife.32532.025

**Figure supplement 1.** Multiple *CEBP* genes are similarly expressed among CD8α⁺ T cell subsets.

DOI: https://doi.org/10.7554/eLife.32532.024

**Figure supplement 1—source data 1.** Data for *Figure 6—figure supplement 1*, mRNA expression from individual experiments.

DOI: https://doi.org/10.7554/eLife.32532.026

for C/EBPδ, and chromatin immunoprecipitation (ChIP) detected binding of C/EBPδ within the 5' flanking regions of these genes, specifically in the MAIT cells (*Figure 7D*).

In analyzing the effects of knocking down C/EBPδ on chemokine receptors, we found that C/EBPδ supported the expression of surface CCR6 and *CCR6* mRNA (*Figure 7E–G*). The 5' flanking region of *CCR6* has multiple predicted binding sites for C/EBPδ (*Figure 7—figure supplement 2*), and just as for *FUT7* and *ST3GAL4*, ChIP showed binding of C/EBPδ to *CCR6* (*Figure 7H*). Again, the effect of knocking down C/EBPδ on the expression of CCR6 was limited to the MAIT cells, concordant with the pattern of C/EBPδ expression.

Based on the absence of any effect of knocking down C/EBPδ on MAIT cell TEM, we anticipated that knocking down C/EBPδ would not affect expression of *CCR2*. As shown in *Figure 7I*, knocking down C/EBPδ did not have a notable effect on the expression of either *CCR2* or *CCR5* in the MAIT cells.

## Discussion

Our data demonstrate heterogeneous, graded expression of the components mediating extravasation into peripheral tissue within populations of human T cells. The abilities to roll, arrest and migrate across an inflamed endothelial cell layer are not all or none and reflect both qualitative and quantitative differences in expression of the essential components, with naïve and MAIT cells occupying extreme positions along a spectrum.

We found that the efficiency of the initial steps of rolling in the flow chamber assays were a function of the levels of surface selectin ligands and sLeˣ, which in turn correlated with expression of *FUT7* and *ST3GAL4*.

FucT-VII is the fucosyltransferase that is critical for synthesis of selectin ligands (*Knibbs et al., 1996*; *Malý et al., 1996*; *Smithson et al., 2001*). In particular, the level of FucT-VII is the determining factor in synthesis of E-selectin ligands (*Knibbs et al., 1996*; *Ley and Kansas, 2004*; *Wagers et al., 1996*). The mouse and/or human genes for FucT-VII are induced in proliferating T cells (*Blander et al., 1999*; *Knibbs et al., 1996*), up-regulated by IL-12 (*Ebel et al., 2015*; *Wagers et al., 1998*; *White et al., 2001*), and suppressed by IL-4 (*Wagers et al., 1998*), related to the activities of T-bet and GATA-3 (*Chen et al., 2006*). Among sialyltransferases, ST3Gal-IV (*Ellies et al., 2002*; *Sperandio et al., 2006*), together with ST3Gal-VI (*Yang et al., 2012*), are the critical enzymes for synthesizing selectin ligands in mice, whereas ST3Gal-IV is the dominant sialyltransferase in synthesizing selectin ligands on myeloid cells in humans (*Mondal et al., 2015*). Like the mouse *Fut7* and/or human *FUT7*, *St3gal4* is induced by T cell activation (*Blander et al., 1999*). However, little is known about how the mouse or human gene for ST3Gal-IV is regulated.

In concert with the progressive increase that we found among the memory-phenotype T cell subsets in levels of selectin ligands and rolling, we found enhanced abilities for firm arrest that could not be explained by differences in integrin expression. Antibody neutralization of the CCR6 ligand, CCL20, resulted in decreased numbers of cells arresting in flow chambers. Our findings are consistent with reports describing the ability of CCR6 to mediate arrest of lymphocytes, including, very

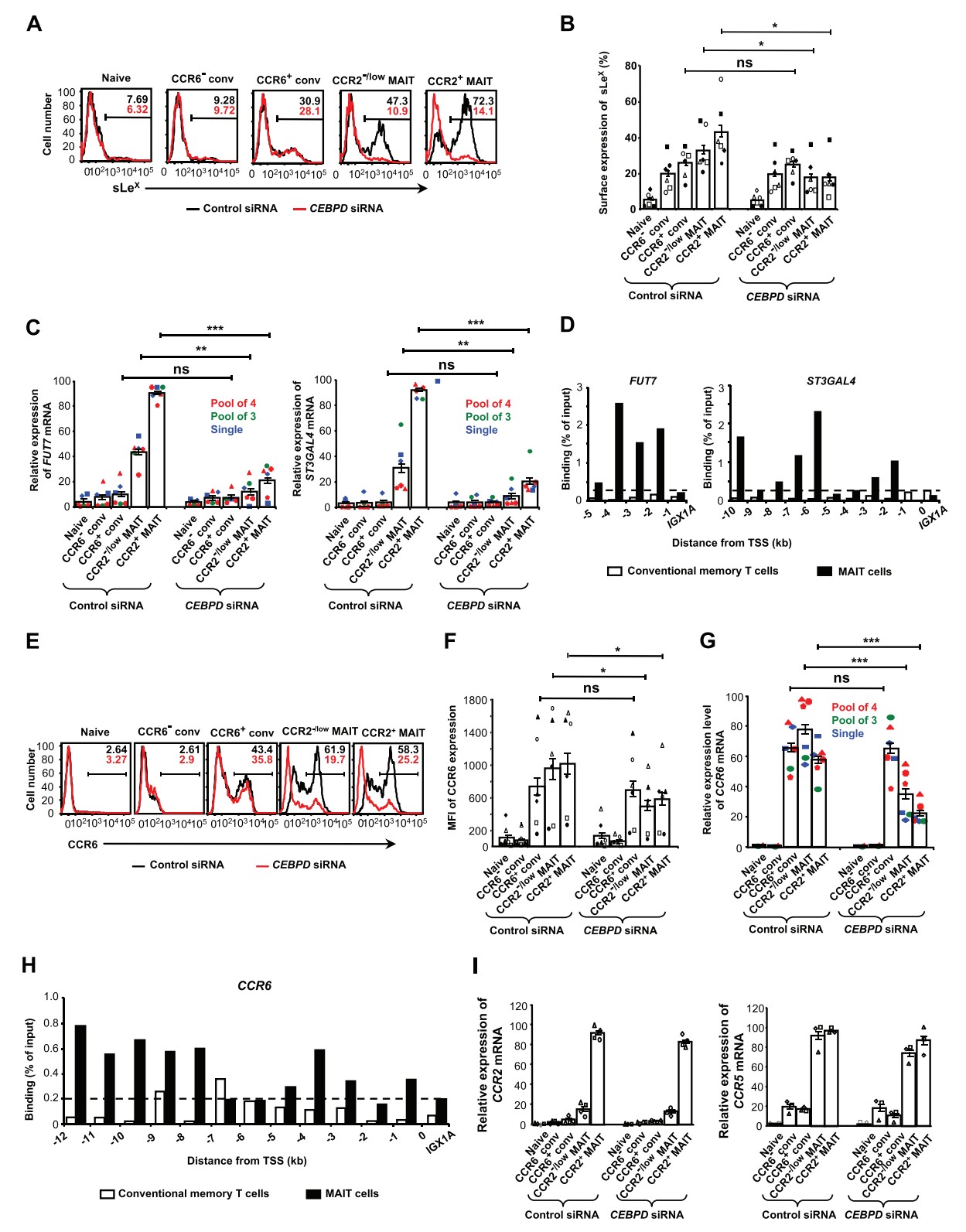

**Figure 7.** C/EBPδ regulates sLe$^X$, *FUT7*, *ST3GAL4* and CCR6/*CCR6* in MAIT cells. (**A**) Expression of sLe$^X$ on CD8α$^+$ T cells after transfections with control or *CEBPD* siRNA. CD8α$^+$ T cells were divided into subsets as in *Figure 1B* and shown in *Figure 2—figure supplement 1*. Percentages of cells staining positive are indicated by the horizontal lines. Unless otherwise noted, experiments in this figure used the pool of four siRNAs, and in all experiments using siRNAs cells were harvested 3–4 days after transfections. (**B**) Expression of sLe$^X$ on T cells from multiple donors treated as in (**A**). (**C**)

*Figure 7 continued on next page*

*Figure 7 continued*

Relative expression of *FUT7* and *ST3GAL4* mRNAs in CD8α⁺ T cell subsets after transfections with control or *CEBPD* siRNAs. Each symbol type shows results from an individual donor. Red and green symbols are from experiments using non-overlapping pools of four or three CEBPD siRNAs, respectively, and blue symbols are from experiments using one of the siRNAs from the pool of three. (D) ChIP analysis of conventional memory-phenotype T cells and MAIT cells using anti-C/EBPδ antibodies and primers for amplifying sequences 5' to the transcription start sites (TSS) for *FUT7* and *ST3GAL4* at 1 kb intervals. Data are expressed as percent of input DNA, and the dashed line indicates 'background' signal based on results for the intergenic region, *IGX1A*. (E) Expression of CCR6 on T cells after transfections with control or *CEBPD* siRNAs. Percentages of cells staining positive are indicated by the horizontal lines. (F) MFIs for CCR6 on T cells treated as in (E). (G) Relative expression of *CCR6* mRNA in CD8α⁺ T cell subsets after transfections with control or *CEBPD* siRNAs as in (C). (H) ChIP analysis as in (D), except using primers for *CCR6*. (I) Relative expression of *CCR2* and *CCR5* mRNAs in CD8α⁺ T cell subsets after transfections with control or CEBPD siRNAs. (A, D, E and H) Data are from one representative of six (A), five (D), left panel), six (D), right panel), six (E), and two (H) donors. (B, C, F, G and I) Bars show means and SEMs, and data are from cells from a total of six (B, C, F, and G), four (I, left panel), or three (I, right panel) donors, each represented by a unique symbol. The p values were calculated using the ratio paired t test. (B, C, F, and G) ns, not significant; *, p<0.05; **, p<0.01; ***, p<0.001.

DOI: https://doi.org/10.7554/eLife.32532.027

The following source data and figure supplements are available for figure 7:

**Source data 1.** Data for *Figure 7B, F* (flow cytometry results from individual experiments), *Figure 7C, G, I* (normalized mRNA expression in cells from individual experiments), *Figure 7D, H* (normalized ChIP-PCR results from individual experiments).
DOI: https://doi.org/10.7554/eLife.32532.030

**Figure supplement 1.** No evidence that C/EBPδ regulates *GCNT1* in MAIT Cells.
DOI: https://doi.org/10.7554/eLife.32532.028

**Figure supplement 1—source data 1.** Data for *Figure 7—figure supplement 2*, predicted C/EBPd binding sequences.
DOI: https://doi.org/10.7554/eLife.32532.031

**Figure supplement 2.** The 5' flanking regions of *FUT7*, *ST3GAL4*, and *CCR6* contain sequences predicted to bind C/EBPδ.
DOI: https://doi.org/10.7554/eLife.32532.029

**Figure supplement 2—source data 2.** Data for *Figure 7—figure supplement 1*, normalized mRNA expression in cells from individual experiments.
DOI: https://doi.org/10.7554/eLife.32532.032

recently, MAIT cells on adhesion-molecule coated plates and/or activated endothelial cells (*Alcaide et al., 2012*; *Campbell et al., 1998*; *Fitzhugh et al., 2000*; *Ghannam et al., 2011*; *Kim et al., 2017*). Of particular interest, the other chemokine receptors on the CD8α⁺ T cells could not substitute for CCR6 in maintaining optimal firm arrest, nor was CCR6/CCL20 necessary for TEM of those cells showing CCR6-independent arrest. On the CD8α⁺ T cells, therefore, CCR6 has a particular role in triggering firm arrest.

CCR6 is notable as the chemokine receptor that is expressed on all T cells that can make IL-17 (*Acosta-Rodriguez et al., 2007*; *Singh et al., 2008*), and CCR6 has been shown to be important for Th17 cell arrest on ICAM-1 (*Alcaide et al., 2012*). In addition to the Th17 cell regulator RORγt, only STAT5A (*Tsuruyama et al., 2016*) and, from our own work, PLZF (*Singh et al., 2015*) have been described as controlling *CCR6*. Based on our experiments using siRNA knockdown and ChIP, we have now identified C/EBPδ as an additional, direct activator of *CCR6*.

The pertussis-toxin resistant arrest that we observed in the memory-phenotype T cell subsets could be viewed as a 'baseline' activity on which inducible integrin activation can be superimposed. Although we are not aware of pertussis-toxin resistant arrest being reported previously for resting memory-phenotype cells, this phenomenon has been described for human effector T cells produced by activation ex vivo (*Shulman et al., 2011*). For those cells, unlike for the CCR6⁺ CD8α⁺ memory-phenotype T cells, there was no pertussis toxin sensitive component to the firm arrest, and their behavior was ascribed to high levels of integrin expression together with constitutive activity of PLC-γ1 (*Shulman et al., 2011*). Non-integrin mediated adhesion is another possibility (*Schneider-Hohendorf et al., 2014*). We have not investigated the mechanism responsible for the pertussis toxin resistant component of arrest by these resting cells, but our data suggest that in this regard the memory-phenotype CD8α⁺ T cells occupy a position intermediate between naïve and activated effector cells.

We found that blocking CCR5 did not affect the T cells' arrest, nor did it prevent cells from undergoing TEM over the 20 min of observation. However, the data indicated that CCR5 functioned to shorten the time between the initiation of crawling and TEM, which, in vivo, would be presumed to speed extravasation. Consistent with our observations, CCR5 has been found not to contribute to leukocyte adhesion, either in flow chambers or in blood vessels (*Diacovo et al., 2005*;

*Shulman et al., 2011*; *Weber et al., 2001*), but nonetheless to have roles in completing the process of extravasation (*Diacovo et al., 2005*).

For CCR2, our data showed a critical and specific role in TEM. We often observed significant TEM not only in the CCR2$^+$ MAIT cells, but also in the CCR2$^{-/low}$ MAIT cells. We presume that the CCR2$^{low}$ MAIT cells within the CCR2$^{-/low}$ MAIT cell samples accounted for the ability of cells in these samples to perform TEM. Because the number of CCR2$^{-/low}$ MAIT cells that we obtained from an individual donor was often inadequate for the flow chamber studies, thereby limiting the number of studies that we could do with these cells, we have not used the CCR2 antagonist to test the possibility that CCR2 was mediating TEM in the CCR2$^{-/low}$ cells.

Although CCR2 has been studied extensively in monocyte biology, relatively little is known about the role of CCR2 on T cells. It is of interest that on monocytes, CCR2 and CCL2 were also described as important for TEM (*Weber et al., 1999*), but not for firm adhesion (*Huo et al., 2001*; *Weber et al., 1999*). Of particular relevance for our studies, CCR2 was reported to be essential for TEM across HUVECs and human dermal microvascular endothelial cells in experiments using a mixed population of activated T cells (*Shulman et al., 2011*). We have previously shown that CCR2 is expressed on a subset of human CCR5$^+$CD4$^+$ T cells that have features of a stable population of highly differentiated long-term memory cells which, based on chemokine receptor expression, TCR activation threshold, and effector cytokine production, are ideally equipped for mediating rapid recall responses in tissue (*Zhang et al., 2010*). In addition, CD4$^+$CCR5$^+$CCR2$^+$ T cells are found in cerebrospinal fluid associated with episodes of relapse in multiple sclerosis, and these cells demonstrate an enhanced ability to migrate across a model of the blood-brain barrier (*Sato et al., 2012*). Recent data in mice have suggested that CCR2 is important for the trafficking into the CNS of a pathological subset of Th17 cells that contributes to chronic and relapsing EAE (*Kara et al., 2015*). Taken together, the data suggest that - unlike for some other chemokine receptors that are associated with individual Th cell lineages - up-regulation of CCR2 (and CCR5) is part of a program directed specifically at conferring the capacity for migration of T cells into tissue.

Given the high degree of ligand/receptor promiscuity, the co-expression of many chemokines during inflammation, the co-expression of multiple chemokine receptors on individual cells, and the shared pathways for chemokine receptor signaling, studies of the chemokine system have often confronted questions of functional redundancy. Our data suggest that CCR6, CCR5, and CCR2 serve sequential and distinct functions in MAIT cell extravasation. From our experiments using anti-CCL20 antibodies, we can conclude that CCL20 is displayed on the surface of the TNF$\alpha$-treated endothelial cells, and from the work of *Shulman et al. (2011)*, we know that CCL2 is sequestered in endothelial cell vesicles and only available to CCR2 within T-cell-endothelial cell synapses. Taken together, these observations suggest that the separate roles for the receptors could be the result of anatomic segregation of their chemokine ligands, which might be a general feature of the chemokine system that limits functional redundancy.

Our interest in understanding the relationships between the expression of chemokine receptors such as CCR6 and CCR2 and the biology of T cell subsets led us to these studies of MAIT cells and their trafficking behavior. Our and/or others' data (*Dusseaux et al., 2011*; *Kim et al., 2017*) showed that the MAIT cells were at the high end of a continuum among CD8$\alpha^+$ T cells as regards expression of selectin ligands and/or multiple tissue-homing chemokine receptors, and at the low end of a continuum as regards expression of CCR7 and CD62L, which are required for entering non-inflamed lymph nodes. The MAIT cells' behavior on activated endothelial cells in the flow chamber assays and in vivo were fully consistent with an enhanced ability to enter inflamed tissue rapidly, without additional phenotypic changes and/or activation within lymphoid organs.

MAIT cells are able to make effector cytokines, and they contain cytotoxic molecules such as perforin, granulysin and granzymes (*Dusseaux et al., 2011*; *Franciszkiewicz et al., 2016*; *Le Bourhis et al., 2013*). MAIT cells are able to produce effector cytokines both in response to TCR activation and directly in response to cytokines such as IL-18 and IL-12 (*Franciszkiewicz et al., 2016*; *Jo et al., 2014*; *Slichter et al., 2016*; *Ussher et al., 2014*). These properties would allow MAIT cells to function locally within the earliest stages of antibacterial, and perhaps antiviral (*Loh et al., 2016*; *van Wilgenburg et al., 2016*) defense. In support of such a role, MAIT cells are decreased in blood of patients with bacterial pneumonias, and can be identified in *M. tuberculosis*-infected lungs (*Le Bourhis et al., 2010*).

We found that in MAIT cells, both *FUT7* and *ST3GAL4*, as well as CCR6, are regulated in part by C/EBPδ, and our ChIP data suggest that *FUT7*, *ST3GAL4*, and *CCR6* are direct targets of C/EBPδ. Knockdown of C/EBPδ resulted in decreased surface expression of both sLe^x and CCR6, although the resulting effects on numbers of rolling cells mediated by selectin ligands was more pronounced than on the specific step of firm arrest mediated in part by CCR6. Since the multiple steps of extravasation occur sequentially and interdependently, C/EBPδ's regulation of rolling and firm arrest significantly affected numbers of cells migrating across the activated endothelial cells in the flow chamber assays. We presume that the effects that we documented in the flow chamber assays were the basis for the decreased entry of MAIT cells transfected with *CEBPD* siRNA into inflamed ears. We have summarized our findings for CCR2-expressing MAIT cells in the cartoon shown in *Video 3*.

Knockdown of C/EBPδ had no independent effect on the final step of MAIT cell TEM in the flow chamber assays, nor, consistent with this finding, on expression of *CCR2* (or *CCR5*). Nonetheless, it is of interest that Yamamoto *et al.* described C/EBP binding sites 3' to the *CCR2* transcriptional start site that were important for *CCR2* promoter activity and that could bind C/EBP proteins, including C/EBPδ, found in nuclear extracts of a human monocytic cell line (*Yamamoto et al., 1999*). These data suggest that our negative results notwithstanding, C/EBPδ might, in fact, have a role in regulating *CCR2* expression in MAIT cells. Regardless, consideration of these data raises the important point that experiments using siRNA knockdown will typically underestimate roles of target genes given that mRNA and protein knockdown are incomplete both at the level of individual cells and across the heterogeneous population of siRNA-transfected cells. Consequently, our findings using siRNA knockdown of C/EBPδ reflect the lower limit of the activities of C/EBPδ in supporting expression of the glycosyltransferases and CCR6, and thereby MAIT cell trafficking.

We attributed the effect of knocking down C/EBPδ on the step of firm arrest to the decrease in expression of CCR6. Nonetheless, it is of interest that studies in mice have shown that ST3Gal-IV sialylates CXCR2 (*Frommhold et al., 2008*), and probably CCR1, and CCR5 (although not CCR2) (*Döring et al., 2014*). For these receptors, sialylation is important for chemokine-dependent activation. There is no evidence for sialylation of CCR6, but it remains possible that in addition to a decrease in expression of CCR6, knockdown of C/EBPδ could have led to diminished CCR6 function through loss of ST3Gal-IV-dependent sialylation. A role for glycosylation in CCR6 function warrants further study.

C/EBPδ is one of six members of the C/EBP family of bZIP transcription factors, each of which contains a basic DNA-binding domain and a 'leucine zipper' dimerization domain (*Ramji and Foka, 2002*). In mice, *Cebpd* is induced in many tissues in response to endotoxin or other inflammatory stimuli, including cytokines such as IL-6 and IL-1, although expression in these contexts is typically transient (*Alam et al., 1992*; *Balamurugan and Sterneck, 2013*; *Ko et al., 2015*). Studies in *Cebpd* knockout mice have demonstrated pleiotropic roles for C/EBPδ, including in multiple models of inflammation and infection (*Yan et al., 2013*; *Chang et al., 2012*; *Duitman et al., 2012*; *Litvak et al., 2009*). C/EBPδ has been implicated in the acute phase response (*Alam et al., 1992*; *Juan et al., 1993*; *Ray and Ray, 1994*) and in the phenotype and function of macrophages (*Balamurugan et al., 2013*; *Chang et al., 2012*; *Duitman et al., 2012*; *Litvak et al., 2009*; *Maitra et al., 2011*; *Yan et al., 2013*; *Litvak et al., 2009*; *Maitra et al., 2011*; *Balamurugan et al., 2013*). Of possible relevance to our own work, C/EBPδ has been suggested to enhance the migration of macrophages into infected lung (*Duitman et al., 2012*). A broader role for C/EBPδ in regulating leukocyte migration in inflammation is suggested by data on the C/EBPδ-mediated induction of chemokines, including the ligands for CCR6 (*Chang et al., 2012*) and CCR2 (*Ko et al., 2014*), and in the transcriptional response of endothelial cells to inflammatory stimuli

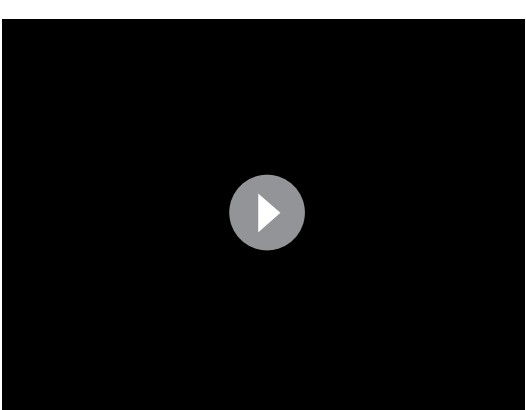

**Video 3.** Summary cartoon, C/EBPδ blocks early steps in MAIT cell trafficking.
DOI: https://doi.org/10.7554/eLife.32532.033

(*Hogan et al., 2017*). In T cells, expression of *CEBPD* was detected in analyzing the transcriptome of human CD8$^+$CD161$^+$ T cells (*Billerbeck et al., 2010*), which would have included MAIT cells, and a bioinformatic analysis of the transcriptome and epigenetic modifications of human CD4$^+$ T cells suggested that C/EBPδ might function as one of the 'master regulators' of T$_{EM}$ differentiation (*Durek et al., 2016*). However, as far as we are aware, functional studies of C/EBPδ in T cells have not been previously reported.

Another transcription factor that has been described with an effect that is analogous but functionally inverse to C/EBPδ is Kruppel-like factor 2 (KLF2), which supports T cell trafficking to lymph nodes versus peripheral tissue by enhancing expression of CD62L and CCR7, and suppressing CXCR3 (*Carlson et al., 2006*; *Preston et al., 2013*). Similarly, SOCS1-mediated inhibition of STATs favors T cell trafficking to lymphoid organs versus peripheral tissue by enhancing expression of CCR7 and suppressing expression of CCR6 and CXCR3 (*Yu et al., 2008*). Based on our data, in contrast to these other factors, C/EBPδ is a positive regulator supporting the tissue-migrating phenotype.

Our study has some clear limitations that affect the generalizability of our conclusions. We studied only human MAIT cells from blood. Our ex vivo experiments used only a single type of endothelial cell (HUVEC) treated with a single pro-inflammatory activator (TNFα), and our in vivo experiments were limited to a simple and artificial model of inflammation at a single tissue site. It is possible that other endothelial cells and/or other activators would differentially affect the abilities of MAIT and non-MAIT cells to arrest and undergo TEM, and thereby reveal activities of other molecules, such as additional transcription factors and chemokine receptors, in these processes. Studies extended to more complex animal models would provide information on the performance of MAIT cells and the molecular mechanisms underlying their trafficking behavior in more biologically relevant models of inflammation. It will be of considerable interest to identify additional factors that control the program for extravasation and the interactions among overlapping networks specifying this activity in coordination with the fates and functions of effector-capable T cells. An understanding of the molecular species regulating the integration of migratory activity and other effector functions of such cells would suggest ways of enhancing or inhibiting T-cell mediated processes in peripheral tissues.

# Materials and methods

## Key resources table

| Reagent type (species) or resource | Designation | Source or reference | Identifiers |
|---|---|---|---|
| Biological sample (human) | Primary human umbilical vein endothelial cell (HUVEC) | ATCC, Manassas, VA | Cat#: PCS-100–013 |
| Biological sample (human) | Human whole blood and elutriated lymphocytes | Department of Transfusion Medicine, Clinical Center, National Institutes of Health | |
| Antibody | Purified-anti-CCL20/MIP-3α (67310) | Minneapolis, MN | Cat#: MAB360 |
| Antibody | Biotin-anti-CCR2 (48607) | R and D Systems | Cat#: FAB151B |
| Antibody | Allophycocyanin-anti-PSGL-1 (688101) | R and D Systems | Cat#: FAB9961R |
| Antibody | FITC-anti-CCR5 (2D7/CCR5) | Franklin Lakes, NJ | Cat#: 561747 |
| Antibody | PE-Cy5-anti-CCR5 (2D7/CCR5) | BD Biosciences | Cat#: 556889 |
| Antibody | Alexa Fluor 647-anti-CCR4 (1G1) | BD Biosciences | Cat#: 557863 |
| Antibody | Alexa Fluor 488-anti-CCR9 (112509) | BD Biosciences | Cat#: 112509 |
| Antibody | PE-anti-CCR10 (1B5) | BD Biosciences | Cat#: 563656 |
| Antibody | FITC-anti-CXCR1 (5A12) | BD Biosciences | Cat#: 555939 |
| Antibody | Allophycocyanin-anti-CXCR2 (6C6) | BD Biosciences | Cat#: 551127 |
| Antibody | Allophycocyanin-anti-CXCR3 (1C6/CXCR3) | BD Biosciences | Cat#: 561324 |

*Continued on next page*

*Continued*

| Reagent type (species) or resource | Designation | Source or reference | Identifiers |
|---|---|---|---|
| Antibody | Allophycocyanin-anti-CXCR4 (12G5) | BD Biosciences | Cat#: 560936 |
| Antibody | Alexa Fluor 647-anti-CXCR5 (RF8B2) | BD Biosciences | Cat#: 558113 |
| Antibody | APC-Cy7-anti-CD8 (SK1) | BD Biosciences | Cat#: 557834 |
| Antibody | Alexa Fluor 700-anti-CD8 (RPA-T8) | BD Biosciences | Cat#:565165 |
| Antibody | PE-Cy5-anti-CD62L (DREG-56) | BD Biosciences | Cat#: 561915 |
| Antibody | FITC-anti-CD62L (DREG-56) | BD Biosciences | Cat#: 555543 |
| Antibody | PE-Cy5-anti-CD45RO (UCHL1) | BD Biosciences | Cat#: 561888 |
| Antibody | Brilliant Violet 605-anti-CD45RO (UCHL1) | BD Biosciences | Cat#: 562641 |
| Antibody | PE-Cy7-anti-CD45RO (UCHL1) | BD Biosciences | Cat#: 337168 |
| Antibody | PE-Cy7-anti-CCR6 (11A9) | BD Biosciences | Cat#: 560620 |
| Antibody | Allophycocyanin-anti-CCR6 (11A9) | BD Biosciences | Cat#: 560619 |
| Antibody | Allophycocyanin-anti-CD161(DX12) | BD Biosciences | Cat#: 550968 |
| Antibody | Non-conjugated anti-sLe$^X$ (CSLEX1) | BD Biosciences | Cat#: 551344 |
| Antibody | PE-conjugated streptavidin | BD Biosciences | Cat#: 349023 |
| Antibody | Alexa Fluor 647-anti-CD31 (390) | San Diego, CA | Cat#: 102416 |
| Antibody | Brilliant Violet 605-anti-CD3 (17A2) | BioLegend | Cat#: 100237 |
| Antibody | FITC-anti-integrin α4 (9F10) | BioLegend | Cat#: 304316 |
| Antibody | Alexa Fluor 647-anti-integrin β1 (TS2/16) | BioLegend | Cat#: 303017 |
| Antibody | Allophycocyanin-anti-integrin β2 (m24) | BioLegend | Cat#: 363410 |
| Antibody | Allophycocyanin-anti-integrin β7 (FIB504) | BioLegend | Cat#: 321208 |
| Antibody | FITC-anti-TCR Vα7.2 (3C10) | BioLegend | Cat#: 351704 |
| Antibody | PE-anti-CXCR6 (K041E5) | BioLegend | Cat#: 356004 |
| Antibody | FITC-anti-CX3CR1 (2A9-1) | BioLegend | Cat#: 341606 |
| Antibody | Allophycocyanin-anti-CD43 (CD43-10G7) | BioLegend | Cat#: 343206 |
| Antibody | Allophycocyanin-anti-CD44 (BJ18) | BioLegend | Cat#: 338806 |
| Antibody | Biotin-anti-IgG Fc (HP6017) | BioLegend | Cat#:409308 |
| Antibody | Anti-CEBPδ (mouse monoclonal) | Dallas, TX | Cat#: sc-135733 |
| Antibody | Anti-CEBPδ | Other | BD 19 |
| Peptide, recombinant protein | Human recombinant TNFα | R and D Systems | Cat#: 210-TA/CF |
| Peptide, recombinant protein | Murine recombinant TNFα | R and D Systems | Cat#: 410-MT/CF |
| Peptide, recombinant protein | Murine recombinant IL-1β | R and D Systems | Cat#: 401 ML-025/CF |
| Peptide, recombinant protein | Human E-selectin Fc chimera | R and D Systems | Cat#: 724-ES |
| Peptide, recombinant protein | Human P-selectin Fc chimera | R and D Systems | Cat#: 137-PS |
| Chemical compound, drug | BMS CCR2 22 | R and D Systems/ Tocris | Cat#: 3129 |
| Chemical compound, drug | Maraviroc | R and D Systems/ Tocris | Cat#: 3756 |

*Continued on next page*

*Continued*

| Reagent type (species) or resource | Designation | Source or reference | Identifiers |
|---|---|---|---|
| Chemical compound, drug | Sialidase (*Vibrio cholera*) | St. Louis, MO | Cat#: N7885-2UN |
| Chemical compound, drug | Pertussis toxin | R and D Systems | Cat#: 3097 |
| Chemical compound, drug | CFSE | Waltham, MA | Cat#: C34554 |
| Chemical compound, drug | CMTPX | Life Technologies | Cat#: C34572 |
| Chemical compound, drug | DAPI | Life Technologies | Cat#: D13076 |
| Sequence-based reagent | CEBPD SMARTpool siRNA | Lafayette, CO | Cat#: L-010453; D-010453–01; D-010453–02; D-01-453-03 |
| Sequence-based reagent | SAMRTpool siRNA control | Dharmacon | Cat#: D-001810-01-05; D-001210–01; |
| Commercial assay or kit | RosetteSep for human CD8 + T cell enrichment | Vancouver, Canada | Cat#: 15063 |
| Commercial assay or kit | qScript cDNA SuperMix | Quanta Biosciences | Cat#: 95048–500 |
| Commercial assay or kit | Perfecta qPCR FastMix UNG ROX | Beverly, MA | Cat#: 95077–012 |
| Commercial assay or kit | RT2 SYBR Green/ROX qPCR Master Mix | Frederick, MD | Cat#: 330522 |
| Commercial assay or kit | Human T Cell Nucleofector Kit | Walkersville, MD | Cat#: VPA-1002 |
| Commercial assay or kit | Magna ChIP A/G kit | Burlington, MA | Cat#: MAGNA0017 |
| Commercial assay or kit | SuperSignal West Pico Chemiluminescent Substrate | Rockford, IL | Cat#: 34080 |
| Software, algorithm | ImageJ | https://imagej.nih.gov/ij/ | |
| Software, algorithm | LAM510 version 4.2 | Wetzlar, Germany | |
| Software, algorithm | Flowjo | Ashland, OR | |
| Software, algorithm | Imaris (Bitplane) | Leica Microsystems | |
| Software, algorithm | Genome Analyzer's Common TF | Ann Arbor, MI | |
| Software, algorithm | Prism | La Jolla, CA | |

## Human cells

HUVECs were cultured according to the supplier's instructions (Promocell, Germany). Human CD8[+] T cells were isolated from elutriated lymphocytes from healthy donors obtained by the Department of Transfusion Medicine, Clinical Center, National Institutes of Health, Bethesda, MD, under a protocol approved by the Institutional Review Board. Informed consent was obtained after explanation of the risks. For use in flow chamber experiments, isolated human CD8[+] T cells were washed and kept overnight in RPMI 1640 (Life Technologies, Waltham, MA) containing 10% FBS (Gemini Bio-Products, West Sacramento, CA), 2 mM L-glutamine, and penicillin-streptomycin (Life Technologies) at 37°C in 5% $CO_2$.

## Mice

C57BL/6J WT mice were purchased from The Jackson Laboratory (Bar Harbor, ME). All mice were used at 8–12 weeks of age. Mice were housed under specific pathogen-free conditions at the National Institutes of Health in an American Association for the Accreditation of Laboratory Animal

Care-approved facility. Animal study protocols were approved by the Animal Care and Use Committee, NIAID, NIH.

## Antibodies and selectin-Fc fusion proteins

All antibodies were against human antigens. Anti–CCR2-biotin (clone 48607), anti–PSGL-1-allophyco-cyanin (688101), human E-selctin Fc chimera and human P-selectin Fc chimera were purchased from R and D Systems, Minneapolis, MN. Anti–CCR5-FITC (2D7), anti–CCR5-PE-Cy5, anti–CCR4-Alexa Fluor 647 (1G1), anti–CCR7-FITC (3D12), anti–CCR9-Alexa Fluor 488 (112509), anti–CCR10-PE (1B5), anti–CXCR1-FITC (5A12), anti–CXCR2-allophycocyanin (6C6), anti–CXCR3-allophycocyanin (1C6), anti-CXCR4-allophycocyanin (12G5), anti–CXCR5-Alexa Fluor 647 (RF8B2), anti–CD8-APC-Cy7 (SK1), anti–CD8-Alexa Fluor 700 (RPA-T8), anti–CD62L-PE-Cy5 (DREG-56), anti–CD62L-FITC, anti–CD45RO-PE-Cy5 (UCHL1), anti–CD45RO-Brilliant Violet 605, anti–CD45RO-PE-Cy7, anti–CCR6-PE-Cy7 (11A9), anti–CCR6-allophycocyanin, non-conjugated anti-sLe$^X$ (CSLEX1), anti-CD161-allophyco-cyanin (DX12), and PE-conjugated streptavidin were purchased from BD Biosciences, Franklin Lakes, NJ. Anti-integrin $\alpha$4-FITC (58XB4), anti-integrin $\beta$1- Alexa Fluor 647 (TS2/16), anti-integrin $\beta$2-allophycocyanin (TS1/18), anti-integrin $\beta$7-allophycocyanin (FIB504), anti-TCRV$_\alpha$7.2-FITC (3C10), anti–CXCR6-PE (K041E5), anti-CX3CR1-FITC (2A9-1), anti–CD43-allophycocyanin (10G7), anti–CD44-allophycocyanin (BJ18) and anti-human IgG Fc-biotin were purchased from BioLegend, San Diego, CA.

## Flow cytometry

For phenotypic analysis of leukocyte subsets, cells were stained in whole blood, in preparations of PBMCs isolated from blood using Ficoll/Hypaque (Amersham Biosciences, United Kingdom), or in preparations of CD8$^+$ T cells purified from elutriated lymphocytes by negative selection using RosetteSep (StemCell Technologies, Canada). For whole blood samples, red cells were removed using Pharm Lyse (BD Biosciences) according to the manufacturer's protocol. For staining other samples, $1 \times 10^5$ cells were suspended in 100 μl of Hanks Balanced Salt Solution (HBSS, Mediatech, Corning, NY) containing 2% FBS. For each sample, cells were incubated with 1 μg of each fluorescent-conjugated primary antibody for 15 min at room temperature (RT), and washed with HBSS/FBS. For cells stained with anti–CCR2-biotin, the cells were incubated with PE-conjugated streptavidin for an additional 15 min at RT. For staining sLe$^X$, the cells were incubated with allophy-cocyanin-conjugated anti mouse IgM (II/41) (BD Biosciences) for an additional 15 min at RT. For staining with E-selectin Fc chimera or P-selectin Fc chimera, the chimeric proteins were first incubated with anti-human IgG Fc-biotin in 100 μl binding buffer (HBSS +5 mM calcium chloride +2 mg/ml BSA) on ice for 10 min before cells were added and incubated on ice for 30 min. After washing with chilled binding buffer, the cells were incubated with streptavidin-PE on ice for 10 min before being washed and analyzed. Staining data were collected on an LSR II cytometer (BD Biosciences). To set gates for defining positive and negative cells in multicolor staining, samples were stained with a mixture of all antibodies save one. Flow cytometry data were analyzed using FlowJo (Ashland, OR).

## Cell sorting

Approximately $1.5 \times 10^8$ CD8$^+$ T cells were isolated from elutriated lymphocytes to approximately 90% purity by negative selection using RosetteSep human CD8$^+$ T cell enrichment cocktail (StemCell Technologies) and incubated with anti–CCR2-biotin and anti–CCR6-PE-Cy7 in HBSS plus 4% FBS for 15 min at RT. Following washing, the cells were stained with streptavidin-PE, anti–CD8-Alexa 700, anti-TCRV$\alpha$7.2-FITC, anti-CD62L-PE-Cy5, and anti–CD45RO-Brilliant Violet 605 for an additional 15 min at RT. The cells were washed and re-suspended in HBSS plus 4% FBS, and cell subsets were isolated to nearly 100% purity using an Aria cytometer (BD Biosciences).

## Analysis of CD8$\alpha^+$ T cell migration under flow

HUVECs were plated at confluence on μ-Slide I 0.4 Luer parallel plate flow chambers (Ibidi, LLC, Germany) which were coated with 50 μg/ml fibronectin (R and D Systems) in PBS, and were stimulated for 18–20 hr with human recombinant TNFα (40 ng/ml (R and D Systems). HUVEC-coated parallel plate flow chambers were assembled with a two-pump system (Harvard Apparatus, Holliston, MA). Sorted T cells were re-suspended at $4 \times 10^5$ cells/ml in perfusion medium

(RPMI 1640 medium containing 2% FBS and 10 mM HEPES). Perfusion of T cells into the flow chambers was performed at 37°C under a force of 0.75 dyn/cm$^2$ for 4 min to allow accumulation of T cells, followed by a constant shear stress of 5 dyn/cm$^2$ for 16 min. Images were acquired at a rate of four frames per second with an integrated fluorescence microscope, Leica AF 6000LX (Leica Microsystems Inc.) with a 20 x DIC objective. For analysis of cell migration, we used Imaris software (Bitplane, South Windsor, CT) to track and categorized cells. We categorized arresting cells as cells that remained stopped on the HUVEC monolayer for more than 10 s under a sheer stress of 5 dyn/cm$^2$; rolling cells as cells that rolled before arresting (whether that arrest had initiated at 0.75 dyn/cm$^2$ or at 5 dyn/cm$^2$), and cells that rolled under a sheer stress of 5 dyn/cm2 but then detached; and transmigrating cells as cells that underwent stepwise darkening under a sheer stress of 5 dyn/cm$^2$.

## T cell trafficking in vivo

Inflammation was induced in the ears of C57BL/6 mice by intradermal injection of murine TNFα (10 μg (R and D Systems) and IL-1β (1 μg (R and D Systems) in 20 μl PBS. Eighteen hours after injection, $1 \times 10^6$ human T cells were injected into the left ventricle, and mice were euthanized 8 min later. Ears were removed, ear sheets were split and cartilage and fat were scraped off. The sheets were then immediately fixed in cold acetone for 20 min for tissue staining or treated in DMEM (Invitrogen, Carlsbad, CA) containing 1 mg/ml DNase I (Sigma-Aldrich St. Louis, MO) and 250 μg/ml Liberase TM (Roche Custombiotech, Indianapolis, IN) for 50 min at 37°C to obtain cell suspensions. Cells were then filtered through a 70 μm nylon mesh and washed prior to counting using the flow cytometer. In some cases, the T cells were labeled with either CMTPX (Life Technologies) or CFSE (Life Technologies) prior to injection.

## Immunofluorescence microscopy

Staining for human CD3, and murine CD31 was done using acetone-fixed ear skin sheets. Skin sheets were blocked for 2 hr at RT with Fc-blocker (BD Biosciences) in PBS containing 4% bovine serum albumin (Sigma-Aldrich). After washing in PBS, skin sheets were incubated with anti-mouse CD31-Alexa Fluor 647 and anti-human CD3-Brilliant Violet 605 antibodies (BioLegend) overnight at 4°C. After washing in PBS, the sections were incubated with DAPI nuclear stain (Invitrogen). Images were acquired using the Carl Zeiss LSM510/Axio Observer confocal microscope and LSM510 version 4.2 software.

## Treatments with sialidase, inhibitors, and neutralizing antibody in flow chamber assays

For removing sialic acid residues, cell-sorted subsets of CD8$^+$ T cells were treated with 0.1 units of sialidase (from *Vibrio cholera*, Sigma-Aldrich) in 1 ml RPMI 1640 (Life Technologies) containing 10% FBS (Gemini Bio-Products) and 2 mM L-glutamine for 2.5 hr at 37°C. For inhibiting $G_{i/o}$ proteins, CD8$^+$ T cells were pre-incubated with pertussis toxin (1 μg/ml (R and D Systems) in RPMI 1640 medium containing 10% FBS and 10 mM HEPES for 3 hr at 37°C. For blocking CCR2 and CCR5, pre-incubation was with BMS CCR2 22 (2 μM (Tocris, Minneapolis, MN) or Maraviroc (10 μM (Tocris), respectively, for 30 min at 37°C and inhibitors were left in the medium throughout the assay. For neutralizing CCL20, HUVEC monolayers in flow chambers were pre-treated for 2 hr at 37°C with 20 μg/ml anti-human CCL20/MIP-3α antibody (c67310; R and D Systems), and antibody was maintained at 10 ng/ml throughout the assay.

## Total RNA isolation and real-time RT-PCR

Subsets of CD8$^+$ T cells were purified by cell sorting as described above. Total cellular RNA was isolated using the TRIzol reagent (Invitrogen). Real-time RT-PCR was performed with 20 ng of RNA as a template, using the qScript cDNA SuperMix for reverse transcription and PerfeCTa qPCR FastMix, UNG, and ROX for PCR (Quanta Biosciences, Beverly, MA). Primer and probe sets (FAM/VIC-labeled) were purchased from Applied Biosystems, Foster City, CA. Results were normalized based on the values for *GAPDH*, detected using TaqMan *GAPDH* control reagents (Applied Biosystems). Real-time qPCR analysis was performed on samples in duplicate using an ABI 7700 Sequence Detection System (Applied Biosystems). For some assays, for cells from each donor, relative levels of

expression, based on values for $2^{-\Delta CT}$, are shown after normalization to the single highest value, which was set to 100. For other assays, values of $2^{-\Delta CT}$ are shown without additional normalization.

## siRNA transfection

Transfections with Amaxa Human T Cell Nucleofector Kit (Lonza, Walkersville, MD) and siRNAs (Dharmacon, Lafayette, CO) were performed following the manufacturers' protocols (Lonza). Five nmol of SMARTpool, SMARTpool control, or individual siRNAs was reconstituted in 250 µl of siRNA buffer, and a total of 10–20 µl was added to a cuvette containing $2-4 \times 10^6$ purified human CD8+ T cells in 100 µl transfection reaction buffer (prepared from Nucleofector Solution and Supplement). The cuvette with cell/siRNA suspension was inserted into the Nucleofector Cuvette Holder and subjected to Nucleofector Program V-024. Five hundred µl of pre-equilibrated culture medium (RPMI 1640 containing 10% FBS, 2 mM L-glutamine, and penicillin-streptomycin) was added to the cuvette, the sample was transferred to the well of 12-well plate, and culture medium was added to a total volume of 2 ml/well. Cells were incubated for 3–4 days at 37°C in 5% $CO_2$. For the in vitro experiments, three experiments used the *CEBPD* SMARTpool containing four siRNAs (catalogue number L-010453) and a SMARTpool siRNA control (catalogue number D-001810-01-05), two experiments used a single *CEBPD* siRNA (catalogue number D-010453–01) that was not part of the *CEBPD* SMARTpool and a control siRNA (catalogue number D-001210–01), and one experiment used this same single *CEBPD* siRNA in a pool with two additional *CEBPD* siRNAs (catalogue numbers D-010453–02 and D-010453–03) not found in the *CEBPD* SMARTpool and a control siRNA (catalogue number D-001210–01). For the experiments using human cells injected into mice, cells were transfected with *CEBPD* SMARTpool containing four siRNAs and a SMARTpool siRNA control.

## Western blotting

Subsets of CD8+T cells were purified by cell sorting as described above and lysed on ice in buffer (30 mM Tris HCl, pH 8.0; 75 mM NaCl; 10% glycerol; and 1% Triton X-100) containing 1:100 proteinase inhibitor cocktail (Cell Signaling, Danvers, MA). Cellular lysates were centrifuged at 12,000 x g for 10 min at 4°C, and supernatants were collected after centrifugation. Protein content was quantified using the Micro BCA protein assay (Pierce, Rockford, lL) according to the manufacturer's guidelines with BSA as a standard. Samples were prepared for SDS-PAGE by boiling at 100°C with 2 x Laemmli sample buffer (Bio-Rad, Hercules, CA) plus 5% β-Mercaptoenthol. A total of 40 µg of cellular proteins and an aliquot of the PageRuler Plus Prestained Protein Ladder (Thermo Scientific, Waltham, MA) were separted by SDS-PAGE in Any kD Mini-PROTEAN TGX Gel (Bio-Rad) at 100 V. After electrophoresis, protein was transferred to an Immun-Blot polyvinylidene difluoride membrane (Bio-Rad) over 1 hr at RT, using a Mini Trans-Blot Cell (Bio-Rad). Following transfer, the membrane was washed in Tris-buffered saline (20 mM Tris, 136 mM NaCl, PH 7.4) with 0.1% Tween 20 (TBST), blocked for 1 hr in TBST/5% nonfat dried milk, incubated overnight at 4°C in the same solution containing 1:10,000 dilution of mouse anti-human C/EBPδ, washed with TBST and incubated at RT with 1:10,000 dilution of HRP-conjugated sheep anti-mouse antibody (GE Healthcare, United Kingdom) in TBST/5% nonfat dried milk for 1 hr, and then washed with TBST at RT. Protein bands were visualized using SuperSignal West Pico Chemiluminescent Substrate (Pierce). Anti-human C/EBPδ was a kind gift from Esta Sterneck, NCI, NIH. Quantification of bands was done using Adobe Photoshop (San Jose, CA) and the ImageJ (NIH) program as described on the following web site: https://www.lukemiller.org/journal/2007/08/quantifying-Western-blots-without.html

## Chromatin immunoprecipitation and quantification of immunoprecipiated DNA

ChIP experiments were performed using Magna ChIP A/G kit from Millipore, Burlington, MA, according to the manufacturer's instruction. Sorted cells were subjected to protein-DNA crosslinking with 1% formaldehyde for 10 min at RT and the reaction was terminated by addition of glycine solution to a final concentration of 125 mM. Cells were re-suspended in cell lysis buffer containing protease inhibitor cocktail. Samples were centrifuged at 2000 rpm for 5 min at 4°C, and the cell pellet was re-suspended in nuclear lysis buffer containing protease inhibitor cocktail. Chromatin was sheared by Bioruptor sonicator (Diagenode, Denville, NJ) to generate DNA fragments between 200 and 1000 base pairs, centrifuged at 13,000 rpm for 10 min at 4°C to pellet debris and diluted 10-fold in

ChIP dilution buffer containing protease inhibitor cocktail. After removing 1% of the sample for analyzing input DNA, the sheared chromatin was incubated at 4°C overnight under rotation with protein A/G magnetic beads and rabbit polyclonal anti-C/EBPδ (Santa Cruz Biotechnology, Dallas, TX) or normal rabbit IgG as a negative control (Millipore), or mouse monoclonal anti-C/EBPδ (C6, Santa Cruz Biotechnology) or normal mouse IgG as a negative control (Millipore). The immunoprecipitates were washed sequentially for 5 min with low-salt immune complex wash buffer, high salt immune complex wash buffer, LiCl immune complex wash buffer and TE buffer. The protein A/G magnetic beads/antibody/chromatin complex was re-suspended in 100 μl of ChIP elution buffer containing 1 μl proteinase K. The crosslinking between DNA and proteins was reversed by incubating the sample at 62°C for 2 hr followed by 95°C for 10 min. DNA was purified by spin column. To analyze promoter regions of *CCR6, FUT7 and ST3GAL4,* we used primers at 1 kb intervals as noted in the legend for *Figure 7*. Quantitative real-time PCR was performed on an Applied Biosystems 7900HT system to determine the relative abundance of target DNA using $RT^2$ SYBR Green/ROX qPCR master mix according to the manufacturer's instructions (SA Biosciences, Frederick, MD). A ChIP PCR primer from *IGX1A* targeting open-reading-frame-free intergenic DNA (SA Biosciences) was used as a negative control. The percent input enrichment was calculated using ChIP PCR array data analysis software (SA Biosciences).

## Computational prediction of C/EBPδ binding sites

The positions of possible transcription factor binding sites (TFBS) in relevant genes were identified in a 20,000 bp region immediately upstream of the transcription start sites (TSS) using the Genomatix Genome Analyzer's Common TF software (Ann Arbor, MI) with default settings. The input sequences were manually obtained from the February 2009 human reference sequence (GRCh37/hg19) in FASTA format. An optimized *CEBPD*-binding weight matrix was constructed based on known binding sites.

## Experimental design and statistical analysis

Sample sizes were chosen based on pilot experiments, and took into consideration the nature of the measurements being made, the magnitudes of differences among groups being compared, and the degrees of donor-to-donor variability. No explicit power analyses were used. In some experiments a minimum of six experiments were performed in order to satisfy the requirements of the Wilcoxon signed rank test. Sample identities were not routinely masked during data acquisition or analysis. Most experimental results were replicated by two investigators. In some cases, only data from one investigator are shown. Otherwise, experiments were excluded only in cases of technical problems that made the results uninformative. We did not define or exclude outliers. Numbers of experimental replicates are noted in the figure legends, and refer to independent experiments and not technical replicates. Given the low abundance of some of the T cells subsets that were studied, most experiments contained a single measurement made for a single sample from a single donor. All statistical tests were performed on directed pairwise comparisons. The tests included two-tailed t tests that were either paired, ratio paired, or unpaired, and the Wilcoxon signed rank test. SEMs are shown for the raw data that were analyzed using the t tests. For some experiments that included control and treated samples, the paired t test was used in place of the ratio paired t test due to the presence of values equal to 0. Tests used for each data set are noted in the figure legends and the source files. No corrections were made for multiple comparisons. Significance is displayed as $*p < 0.05$; $**p < 0.01$; $***p < 0.001$. Statistical analyses were done using Prism (GraphPad, La Jolla, CA).

## Acknowledgements

We thank Shikha Sharan and Esta Sterneck for advice and reagents for detecting C/EBPδ, Hye-lin Ha for advice about immunohistochemistry of skin, members of the Research Technologies Branch, NIAID, for their help with cell sorting, and Philip Murphy for critical reading of the manuscript. The study was supported by the Intramural Research Program of NIAID, NIH.

## Additional information

### Funding

| Funder | Grant reference number | Author |
|---|---|---|
| National Institutes of Health | Intramural Research Program | Joshua M Farber |

The funders had no role in study design, data collection and interpretation, or the decision to submit the work for publication. NIAID approved submission per standard Institute procedures.

### Author contributions

Chang Hoon Lee, Formal analysis, Investigation, Visualization, Writing—original draft; Hongwei H Zhang, Formal analysis, Validation, Investigation, Visualization; Satya P Singh, Validation, Investigation; Lily Koo, Investigation, Visualization; Juraj Kabat, Hsinyi Tsang, Software, Visualization; Tej Pratap Singh, Investigation; Joshua M Farber, Conceptualization, Supervision, Funding acquisition, Project administration, Writing—review and editing

### Author ORCIDs

Chang Hoon Lee (iD) https://orcid.org/0000-0001-8953-9069
Hongwei H Zhang (iD) http://orcid.org/0000-0001-7298-8802
Joshua M Farber (iD) http://orcid.org/0000-0003-3224-0378

### Ethics

Human subjects: Human blood cells were obtained by the Department of Transfusion Medicine, Clinical Center, National Institutes of Health, Bethesda, MD, under protocol 99-CC-0168 approved by the Institutional Review Board. Informed consent was obtained after explanation of the risks.
Animal experimentation: Mice were housed under specific pathogen free conditions at the National Institutes of Health in an American Association for the Accreditation of Laboratory Animal Care-approved facility. Animals were studied under protocol LMI-13 as approved by the Animal Care and Use Committee, NIAID, NIH.

### Decision letter and Author response

Decision letter https://doi.org/10.7554/eLife.32532.036
Author response https://doi.org/10.7554/eLife.32532.037

## Additional files

### Supplementary files

• Transparent reporting form
DOI: https://doi.org/10.7554/eLife.32532.034

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
