## [Decision Letter]

Thank you for submitting your article "A program for transendothelial migration of human MAIT cells controlled by the transcription factor C/EBPd" for consideration by *eLife*. Your article has been reviewed by two peer reviewers, and the evaluation has been overseen by a Reviewing Editor and Tadatsugu Taniguchi as the Senior Editor. The following individuals involved in review of your submission have agreed to reveal their identity: Olivier Lantz (Reviewer #1); Andy Luster (Reviewer #2).

Normally, the reviewers would have discussed the reviews with one another in order to provide a consensus view for the Reviewing Editor to draft a decision to help you prepare a revised submission. In this case, given the favorable reviews, we did not try to consolidate the reviews, since this would have added some time before the authors would have received the decision.

Summary:

In this manuscript, Lee et al. investigate the biology of human mucosal-associated invariant T (MAIT) cells with a focus on the mechanisms regulating their trafficking across inflamed endothelium. Our knowledge of MAIT cell biology in general, and their trafficking patterns specifically, is limited. Given the evidence that MAIT cells contribute to the early response to bacterial infections as well as autoimmune diseases, identifying novel mechanisms regulating MAIT cell trafficking would represent an important contribution.

Essential revisions:

The manuscript was reviewed by two experts, one on MAIT cells, and the other on chemokine biology. Both reviews were favorable but there were a few suggestions for improving the manuscript, as described in detail in the specific comments, attached below.

*Reviewer #1:*

This work studies how human MAIT cells as compared to naive or conventional CD8 T cells interact with the endothelial cells. The authors show that MAIT cells identified as CD161hi Va7.2+ in the CD8a+ subset, especially those that express high levels of CCR2 are the most efficient to arrest, role and transmigrate. The authors use HUVEC as a source of endothelial cells as well as inflamed mouse ear endothelium. The authors then analyse the mechanisms involved to show that MAIT cells express the highest levels of selectin ligands with high expression of the enzymes (FUT7, ST3GAL4) required to make the selectin ligands. MAIT express high level of CCR2, CCR5 and CCR6. Blocking experiments indicate that CCR2 and CCR6 play a non-redundant role in diapedesis whereas CCR5 accelerates transmigration. The authors then show that the transcription factor CEBPD controls the receptor for the integrins involved in the rolling step of diapedesis since siRNA mediated inhibition of this factor decreased mostly the initial step of diapedesis (rolling). Interestingly, inhibition CEBPD decreased mRNA levels of FUT7, ST3GAL4 and CCR6 in MAIT cells but not in other memory T cells.

The study is well designed and the experiments are clear and demonstrative with a high number of independent experiments or donors allowing firm conclusions. This work adds new information in the field further emphasizing the particular features of MAIT cells allowing them to efficiently transmigrate to tissue. The mechanisms are well analyzed. Although rather long, the Introduction and the Discussion are interesting and well written.

*Reviewer #2:*

In this manuscript, Lee et al. investigate the biology of human mucosal-associated invariant T (MAIT) cells with a focus on the mechanisms regulating their trafficking across inflamed endothelium. Our knowledge of MAIT cell biology in general, and their trafficking patterns specifically, is limited. Given the evidence that MAIT cells contribute to the early response to bacterial infections as well as autoimmune diseases, identifying novel mechanisms regulating MAIT cell trafficking would represent an important contribution.

The Farber group previously defined a subset of human memory CD4^+^ T cells that were programmed for a rapid recall response and expressed the chemokine receptor CCR2. In the present study, they have extended their findings and show that the majority of human blood-derived CD8^+^ T cells expressing CCR2 are MAIT cells. They further characterize the homing molecules on MAIT cells utilizing flow cytometry and transcriptional profiling and go on to demonstrate non-redundant roles for selectin ligands and the chemokine receptors CCR2, CCR5, and CCR6. Finally, the authors demonstrate that the transcription factor C/EBPδ is highly expressed in MAIT cells compared to conventional memory CD8^+^ T cells, and directly promotes the expression of the glycosylation enzymes FUT7 and ST3GAL4, and to a lesser extent CCR6. C/EBPδ expression was shown to enhance MAIT cell rolling, arrest, and transendothelial migration in vitro as well as extravasation in vivo. However, comparing arrested/rolling and TEM/arrested cells suggests that the predominant effect of C/EBPδ is on selectin-mediated rolling. Interestingly, C/EBPδ knockdown did not affect the expression of CCR2, which was required for efficient TEM, suggesting that C/EBPδ-dependent and independent pathways regulate expression of important homing molecules in MAIT cells.

To my knowledge, there have been few transcription factors identified that regulate lymphocyte homing programs. KLF2 is a notable exception and controls the expression of genes involved in lymph node homing, such as S1P1, CD62L and β7 integrin. The identification of a transcription factor that regulates a genetic program governing lymphocyte trafficking to the inflamed periphery would therefore be of interest. In this regard, identifying C/EBPδ as a regulator of MAIT cell trafficking to the inflamed periphery is a novel contribution. Furthermore, while CCR6 expression and function have been investigated in MAIT cells (Dusseaux et al., 2011), the finding that CCR6 partially mediated MAIT cell arrest while CCR2 solely mediated their TEM on TNF-activated HUVECs are novel findings.

However, the title of the paper was a bit misleading as the two major findings of the study seem unrelated; C/EBPδ regulated selectin-ligand expression and rolling, whereas C/EBPδ did not regulate CCR2 expression and TEM. C/EBPδ also only had a modest effect on CCR6 expression and MAIT cell arrest. Thus, the data seems to support the conclusion that C/EBPδ is a regulator of MAIT cell trafficking to the periphery by controlling selectin ligand expression more than being a master regulator of a genetic program that controls multiple trafficking genes collectively required for lymphocyte trafficking to the inflamed periphery. The manuscript begins by describing CCR2 expression on MAIT cells then ends with C/EBPδ. The lack of relationship between these two findings weakens the manuscript given the way the it was written. A re-organization and re-write with a different emphasis might be helpful. Given that MAIT cells demonstrate enhanced trafficking across inflamed endothelium compared to other CD8^+^ lymphocytes, perhaps the authors could use this as a starting point and then describe two independent pathways controlling this process: 1) C/EBPδ-dependent regulation of selectin ligands that promotes rolling; and 2) a CCR2 pathway that promotes TEM (C/EBPδ-independent).

---

## [Author Response]

Essential revisions:The manuscript was reviewed by two experts, one on MAIT cells, and the other on chemokine biology. Both reviews were favorable but there were a few suggestions for improving the manuscript, as described in detail in the specific comments, attached below.Reviewer #2:[…] However, the title of the paper was a bit misleading as the two major findings of the study seem unrelated; C/EBPδ regulated selectin-ligand expression and rolling, whereas C/EBPδ did not regulate CCR2 expression and TEM. C/EBPδ also only had a modest effect on CCR6 expression and MAIT cell arrest. Thus, the data seems to support the conclusion that C/EBPδ is a regulator of MAIT cell trafficking to the periphery by controlling selectin ligand expression more than being a master regulator of a genetic program that controls multiple trafficking genes collectively required for lymphocyte trafficking to the inflamed periphery. The manuscript begins by describing CCR2 expression on MAIT cells then ends with C/EBPδ. The lack of relationship between these two findings weakens the manuscript given the way the it was written. A re-organization and re-write with a different emphasis might be helpful. Given that MAIT cells demonstrate enhanced trafficking across inflamed endothelium compared to other CD8^+^ lymphocytes, perhaps the authors could use this as a starting point and then describe two independent pathways controlling this process: 1) C/EBPδ-dependent regulation of selectin ligands that promotes rolling; and 2) a CCR2 pathway that promotes TEM (C/EBPδ-independent).

I very much appreciate Dr. Luster’s thoughtful critique. Although I chose not to re-organize the manuscript in fundamental ways (please see below), I believe that I have made other modifications that should help to address Dr. Luster’s principal concerns. Dr. Luster found the organization of the manuscript confusing in that it opened with a focus on CCR2-expressing cells and then shifted to C/EBPδ, which did not have a demonstrated role in regulating CCR2. I believe that adding to the confusion was the inconsistent use of the term “transendothelial migration”. I have now limited the use of this term to refer specifically to the step of crossing the endothelial cells, and because this step was not affected by knockdown of C/EBPδ, the title and the Abstract have been changed. I have made additional changes to the Abstract and at the end of the Introduction in order to make clear at the outset that our data support roles for C/EBPδ in expression of selectin ligands and CCR6, and consequently in MAIT cell rolling and arrest, but not in the step of TEM, which depends on CCR2. I have also moved the data showing the absence of an effect on expression of CCR2 (and CCR5) from the supplemental data to Figure 7 believe that these modifications will provide the reader with the proper orientation and allay confusion as he/she follows the presentation of the data.

Dr. Luster was not enthusiastic about the claim that C/EBPδ regulated a broad program for extravasation, which was referred to in the original title. I believe that his reluctance was based in part on the rather modest impact of C/EBPδ knockdown on the specific step of arrest, and his conclusion that the major role of C/EBPδ was in supporting the expression of selectin ligands and the step of MAIT cell rolling. In order to satisfy this concern, the revised title does not make such a claim, and the Abstract and other text make clear that C/EBPδ is only one part of the regulation of the extravasation program. In addition, I make the point in the revised Discussion that knocking down C/EBPδ had a more pronounced effect on cell rolling than on arrest.

Nonetheless, I believe that our data do show an effect of knocking down C/EBPδ on expression of CCR6 and an effect on arrest in the flow chamber assays, in addition to the effects on the glycosyltransferases and selectin-mediated rolling. We also show binding of C/EBPδ to *CCR6* using ChIP. I would emphasize, as I do in the revised Discussion (please see below) that knockdown experiments are likely to underestimate the magnitudes of the activities of the target genes/proteins, and that our flow chamber assays cannot replicate the myriad of conditions under which the MAIT cells traffic in vivo where arrest may be more (or less) CCR6-dependent.

I believe that it is of significant interest that C/EBPδ regulates genes whose proteins show disparate activities, yet which all contribute to steps in extravasation. Efficient extravasation into inflamed tissue requires the cooperative activities of many proteins, which suggests an underlying coordinated program of gene regulation. Our data on C/EBPδ support the existence of such a program in MAIT cells.

I carefully considered Dr. Luster’s most consequential suggestion for reorganizing the manuscript by starting with observations on MAIT cells generally and then separating the presentation into C/EBPδ-dependent and C/EBPδ-independent contributors to migration. Nonetheless, I chose not to revise the manuscript along these lines. All the trafficking data were collected on cells separated into the designated subsets, including separations based on expression of CCR2, as shown in Figure 1 and Figure 2—figure supplement 1, and the data on all steps in trafficking, including those that we found affected and unaffected by knockdown of C/EBPδ, needed to be shown together. As a result, I believe that it would be difficult to separate the presentation of the data based on C/EBPδ-dependent rolling and arrest and C/EBPδ-independent TEM. The modifications described above should make it clear throughout the manuscript that we found steps in MAIT cells trafficking that were and steps that were not sensitive to knockdown of C/EBPδ.

Moreover, I was reluctant to use C/EBPδ-dependent and C/EBPδ-independent pathways as a central organizing principal given that negative results using siRNA knockdown are not definitive. I now discuss this limitation of the siRNA data in the Discussion along with data from the literature suggesting that the C/EBP transcription factors, including C/EBPδ, may activate *CCR2* in monocytes. These data provide an additional basis for caution in relying on the C/EBPδ knockdown data for the conclusive identification of C/EBPδ-independent regulation. Even though our knockdown of C/EBPδ was sufficient to affect expression of the glycosyltransferase genes and *CCR6*, there might be target genes (?*CCR2*) for which that degree of knockdown was not adequate to see an effect. I believe that the manuscript is organized logically, investigating the mechanisms underlying the trafficking steps in the sequence in which they occur in blood vessels, followed by the additional data on those steps and genes that were sensitive or insensitive to knockdown of C/EBPδ.